# SeMoBridge: Semantic Modality Bridge for Efficient Few-Shot Adaptation of CLIP

## Abstract

While Contrastive Language-Image Pretraining (CLIP) excels at zero-shot tasks by aligning image and text embeddings, its performance in few-shot classification is hindered by a critical limitation: *intra-modal misalignment*. This issue, caused by a persistent *modality gap* and CLIP's exclusively inter-modal training objective, leaves the embedding spaces uncalibrated, making direct image-to-image comparisons unreliable. Existing methods attempt to address this by refining similarity logits or by computationally expensive per-sample optimization. To overcome these challenges, we introduce SeMoBridge, a lightweight yet powerful approach that directly addresses the misalignment. Our method maps images into the text modality, while keeping their semantic content intact through what we call a *Semantic Modality Bridge*. SeMoBridge is closed-form and can optionally be trained through multi-modal supervision, combining image and text-alignment losses to optimize the projection. Experiments show that the trained version, SeMoBridge-T, requires only a fraction of the training time while overall outperforming other methods, particularly in low-data scenarios (1, 2, and 4 shots).

## 1 Introduction

Contrastive Language-Image Pretraining (CLIP) (Radford et al., 2021) consists of a vision encoder and a text encoder that are jointly trained to map images and text into a shared embedding space. By leveraging large-scale image-text pairs and optimizing a contrastive objective, CLIP achieves strong inter-modal alignment and remarkable generalization capability. Owing to these properties, CLIP has been widely adopted for downstream tasks such as zero-shot and few-shot classification.

In few-shot classification, a query image must be matched against a small set of labeled examples, which requires accurate image-to-image comparison. Since this is a comparison within the same modality, it can be viewed as an *intra-modal* comparison and thus relies on well-calibrated intra-modal alignment.

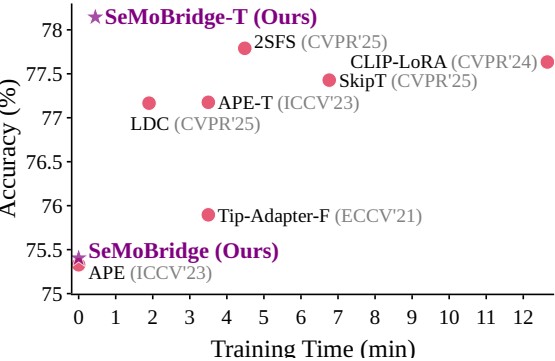

Figure 1: Comparison of average Accuracy against Training Time of few-shot image classification methods on 11 datasets. Our proposed trained SeMoBridge-T achieves better accuracy using only a fraction of the time.

However, CLIP embeddings inherently suffer from a *modality gap* (Liang et al., 2022), i.e., a separation between image and text modalities. This separation, present from initialization, is not resolved by CLIP's training. Instead, the contrastive objective's focus on pulling paired samples together across the gap leaves the internal semantic structure of each modality uncalibrated. As a consequence, as shown in Figure 2, a query image of a dog can be mistakenly placed closer to the cat few-shot centroid than to its correct dog counterpart ($d_{cat} < d_{dog}$), resulting in misclassification.

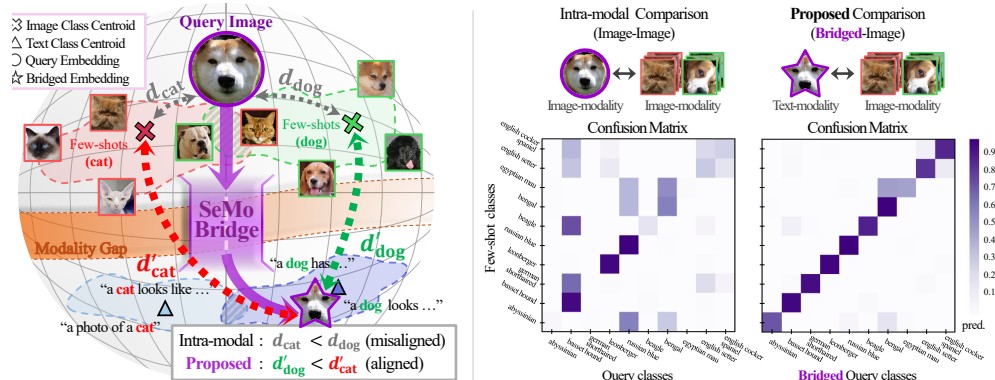

Figure 2: **Left:** Illustration of the modality gap, intra-modal misalignment, and our proposed *Semantic Modality Bridge* (SeMoBridge). Due to intra-modal misalignment, query images can be embedded closer to the wrong class. SeMoBridge addresses this by applying a single unified projection that maps image embeddings into the text modality, preserving their semantics and enabling more accurate comparison. **Right:** Confusion matrices on a subset of 10 classes from the Oxford-Pets dataset, comparing intra-modal and our bridged inter-modal approach. Each matrix shows how query images are classified with respect to the few-shot support classes. SeMoBridge substantially reduces class confusion by enabling more reliable comparisons.

Existing methods tackle this problem from two main directions. Some, such as Tip-X (Udandarao et al., 2023) and APE (Zhu et al., 2023) avoid direct image-to-image comparisons altogether, relying on indirect similarity measures via text prompts. While effective to some extent, this limits the ability to capture fine-grained visual details. In contrast, Cross the Gap (Mistretta et al., 2025) directly addresses this issue by mapping images to the text modality but requires computationally expensive, per-sample optimization.

This leads us to ask our work's central question: *Can we design a method that overcomes intra-modal misalignment without the high computational cost of per-sample optimization?*

In this paper, we answer this by introducing SeMoBridge, a lightweight and efficient *Semantic Modality Bridge*. It utilizes the pre-trained semantic structure of CLIP to create a single unified projection that is applicable to all inputs and allows direct comparison of images. The advantage of our method is illustrated in Figure 2 (left): Applying SeMoBridge enables aligned inter-modal comparisons between the bridged query image and few-shots ($d'_{\text{dog}} < d'_{\text{cat}}$). Figure 2 (right): The confusion matrices confirm this effect. Intra-modal comparisons often misclassify query images, whereas SeMoBridge reduces such confusion through bridged inter-modal comparison.

Although SeMoBridge is designed to be training-free, it can be efficiently fine-tuned through multi-modal supervised training that improves semantic alignment. By updating only the lightweight bridge while keeping CLIP frozen, our method achieves a very low training cost, as shown in Figure 1. Extensive experiments across diverse benchmarks confirm that SeMoBridge achieves state-of-the-art few-shot performance, especially in low-data scenarios.

We summarize the contributions of our work as follows:

- SeMoBridge, a **lightweight, training-free Semantic Modality Bridge for efficient few-shot adaptation of CLIP** that minimizes compute by avoiding per-sample optimization.

- A **novel multi-modal supervision strategy** combining image and text-alignment losses ensuring bridged embeddings keep semantic knowledge from both modalities, while avoiding backpropagation through CLIP's encoders.

- Extensive experiments that show **SeMoBridge outperforms existing methods in few-shot learning with significantly less training time**, achieving higher accuracy and better generalization, especially in very low-shot scenarios.

## 2 RELATED WORK

**Challenges in CLIP Few-shot Adaptation.**   Vision-language models such as CLIP (Radford et al., 2021) have demonstrated strong performance in zero-shot and few-shot classification by embedding images and texts into a shared semantic space. To leverage this, numerous methods have been proposed to adapt CLIP to few-shot settings without modifying its pretrained backbone. However, CLIP few-shot adaptation is challenged by intra-modal misalignment, which arises from the inherent modality gap in CLIP. This misalignment makes direct image-to-image comparisons unreliable, motivating the need for more robust adaptation strategies that address this problem.

**Types of CLIP Few-shot Adaptation.**   Several approaches operate only at the prediction logit level, based on Tip-Adapter (Zhang et al., 2021). Tip-X (Udandarao et al., 2023) addresses intra-modal misalignment by bypassing direct image-to-image comparisons. Instead, it maps both query and few-shot images into CLIP's logit space by computing similarity distributions to a set of class text prompts. These distributions are then compared using KL-divergence, forming an indirect measure of similarity between images. Adaptive Prior rEfinement (APE) (Zhu et al., 2023) refines CLIP embeddings through feature selection and computes trilateral affinities among query, few-shot, and text features, leading to more semantically accurate representations and robust predictions. Unlike methods that rely on indirect similarity comparisons between query and few-shot embeddings, Logit DeConfusion (LDC) (Li et al., 2025) introduces adapter modules that leverage the few-shot set to learn class-level confusion patterns in CLIP, and applies corrections to improve classification.

While effective, these approaches operate only at the output logit level, and thus cannot fully leverage the inter-modal semantic priors in CLIP. *We argue that a better adaptation can be achieved by operating within the embedding space itself.*

**Optimization-based Modality Inversion.**   Overcoming the limitations of previous approaches, recent work (Mistretta et al., 2025) introduces a direct embedding transformation method based on modality inversion. They propose Optimization-based Textual Inversion (OTI), which learns a pseudo-text token from a given image embedding, and Optimization-based Visual Inversion (OVI), which is the reverse. While this approach provides a solution for intra-modal misalignment, it requires iterative optimization at inference for every image or text sample, which increases computational cost and limits flexibility.

**Closed-form Modality Inversion.**   SD-IPC (Ding et al., 2023) proposes a closed-form projection method originally developed for converting image embeddings into the prompt embedding space of Stable Diffusion (Rombach et al., 2022). Unlike OTI/OVI, which employ iterative optimization, SD-IPC leverages the pre-trained alignment between CLIP's image and text embeddings. It enables efficient closed-form inversion without iterative optimization per-sample. This provides a lightweight and general-purpose mechanism for modality inversion, although it is not designed for classification.

**Relations to Our Approach.**   Unlike existing approaches, our proposed SeMoBridge is the first to address intra-modal misalignment by fully leveraging CLIP's inter-modal semantic priors, while remaining efficient and closed-form. In contrast to methods which operate only with similarity logit refinement, SeMoBridge bridges the modality gap by mapping image embeddings into the text space, enabling more reliable inter-modal comparisons. Different from OTI/OVI, that require expensive per-sample optimization at inference, our method eliminates this overhead through a single shared projection that generalizes across all samples.

## 3 METHODOLOGY

### 3.1 PRELIMINARIES.

**CLIP Review.**   We utilize the pre-trained CLIP model (Radford et al., 2021), which maps images and texts into a shared $d$-dimensional embedding space. Given an image $x$ and a corresponding caption $t$ (e.g., "a shiba inu smiling into the camera"), their embeddings are computed as follows:

$$\mathbf{x}_{\text{enc}} = \text{Enc}_{\text{img}}(x) \in \mathbb{R}^{d_i}, \quad \mathbf{x}_{\text{img}} = \mathbf{W}_{\text{img}}(\mathbf{x}_{\text{enc}}) \in \mathbb{R}^{d},$$

$$\mathbf{t}_{\text{eos}} = \text{EOS}(\text{Enc}_{\text{txt}}(t)) \in \mathbb{R}^{d_t}, \quad \mathbf{t}_{\text{txt}} = \mathbf{W}_{\text{txt}}(\mathbf{t}_{\text{eos}}) \in \mathbb{R}^d,$$

where $\text{Enc}_{\text{img}}$ is the image encoder and $\text{Enc}_{\text{txt}}$ the text encoder. $\text{EOS}(\cdot)$ extracts the end-of-sequence (EOS) token from the text encoder's output, which contains the semantic summary of the text input. Finally, both images and texts are projected to $\mathbf{x}_{\text{img}}$ and $\mathbf{t}_{\text{txt}}$ through $\mathbf{W}_{\text{img}}$ and $\mathbf{W}_{\text{txt}}$, respectively, and then aligned in the shared space through contrastive training.

**Few-shot classification problem.** CLIP embeds transferable representations that enable both zero-shot and few-shot learning across diverse visual concepts. Our goal is to predict the class label $y_q \in \{1, ..., C\}$ of a query image $x_q$ by leveraging the given few-shot set $\mathcal{D} = \{(x_i, y_i)\}_{i=1}^{C \times K}$, of $K$ shots for each class, totaling $C \times K$ images. $\mathbf{L} \in \mathbb{R}^{CK \times C}$ denotes the one-hot encoded labels for the few-shot set.

An intuitive approach is to embed both the query image $\mathbf{f}_{\text{img}} \in \mathbb{R}^d$ and few-shot images $\mathbf{F}_{\text{img}} \in \mathbb{R}^{C \times K \times d}$ with CLIP, and then compute class-wise similarities $\mathbf{f}_{\text{img}}\mathbf{F}_{\text{img}}^{\top}$ for classification. However, due to intra-modal misalignment, image-image comparisons are often unreliable. As shown in Figure 2, they can be noisy and fail to reflect true semantic relationships, highlighting the need for a more robust solution.

## 3.2 SEMANTIC MODALITY BRIDGE.

We address this challenge by converting intra-modal comparisons into robust inter-modal ones, leveraging the strong image-text alignment that CLIP was trained to learn. Our idea is to adapt this alignment for few-shot classification by producing a text-like bridged embedding from an image that allows more reliable comparisons within CLIP's shared space.

To achieve this, we build on SD-IPC (Ding et al., 2023), which introduced a method for deriving a "pseudo" End-of-Sequence (EOS) token from an image embedding that preserves its semantics. While originally proposed for generating prompts in text-to-image models such as Stable Diffusion (Rombach et al., 2022), we repurpose this mechanism for few-shot classification. Formally, we first derive a pseudo-EOS token $\hat{\mathbf{f}}_{\text{eos}}$ using SD-IPC's approach. Then we map it through CLIP's text projection layer $\mathbf{W}_{\text{txt}}$ to obtain our final bridged embedding $\hat{\mathbf{f}}_{\text{txt}}$, which can be directly and reliably compared with image embeddings in CLIP's shared space.

This process is justified by CLIP's training objective, which explicitly maximizes the cosine similarity between paired image and text embeddings. This forces their vector representations to *point in the same direction* within the shared space. Based on this, we can assume that the normalized vectors of an image embedding $\mathbf{f}_{\text{img}}$ and its corresponding (but unknown) text embedding $\hat{\mathbf{f}}_{\text{txt}}$ are approximately equal:

$$\frac{\mathbf{f}_{\text{img}}}{\|\mathbf{f}_{\text{img}}\|} \approx \frac{\hat{\mathbf{f}}_{\text{txt}}}{\|\hat{\mathbf{f}}_{\text{txt}}\|}, \quad \text{where } \hat{\mathbf{f}}_{\text{txt}} = \mathbf{W}_{\text{txt}}\hat{\mathbf{f}}_{\text{eos}}. \tag{1}$$

With this approximation, we can estimate the unknown pseudo-EOS token $\hat{\mathbf{f}}_{\text{eos}}$. The idea is to back-project the pseudo text embedding $\hat{\mathbf{f}}_{\text{txt}}$ through the text projection matrix using its Moore–Penrose pseudo-inverse $\mathbf{W}_{\text{txt}}^{+}$ (Penrose, 1955). Since Eq. 1 implies that $\mathbf{f}_{\text{img}}$'s direction is aligned with $\hat{\mathbf{f}}_{\text{txt}}$, we can substitute $\mathbf{f}_{\text{img}}$ in its place.

$$\hat{\mathbf{f}}_{\text{eos}} \approx \frac{\|\hat{\mathbf{f}}_{\text{eos}}\|}{\|\mathbf{W}_{\text{txt}}^{+}\mathbf{f}_{\text{img}}\|}\mathbf{W}_{\text{txt}}^{+}\mathbf{f}_{\text{img}} \approx \frac{\|\mathbf{T}_{\text{eos}}\|}{\|\mathbf{W}_{\text{txt}}^{+}\mathbf{f}_{\text{img}}\|}\mathbf{W}_{\text{txt}}^{+}\mathbf{f}_{\text{img}}. \tag{2}$$

After the inverse projection, *the magnitude (norm) may not match that of a real EOS token* $\hat{\mathbf{f}}_{\text{eos}}$. To correct for this, we rescale $\|\mathbf{W}_{\text{txt}}^{+}\mathbf{f}_{\text{img}}\|$ so that it matches genuine EOS tokens. However, the true norm $\|\hat{\mathbf{f}}_{\text{eos}}\|$ is unknown, we approximate it as the average EOS norm across all class descriptions. $\|\mathbf{T}_{\text{eos}}\|$, which is computed as $\frac{1}{CK}\sum_{c=1}^{C}\sum_{k=1}^{K}\|\mathbf{T}_{\text{eos}}^{c,k}\|$.

Finally, we project it into the shared space to get the final bridged embedding:

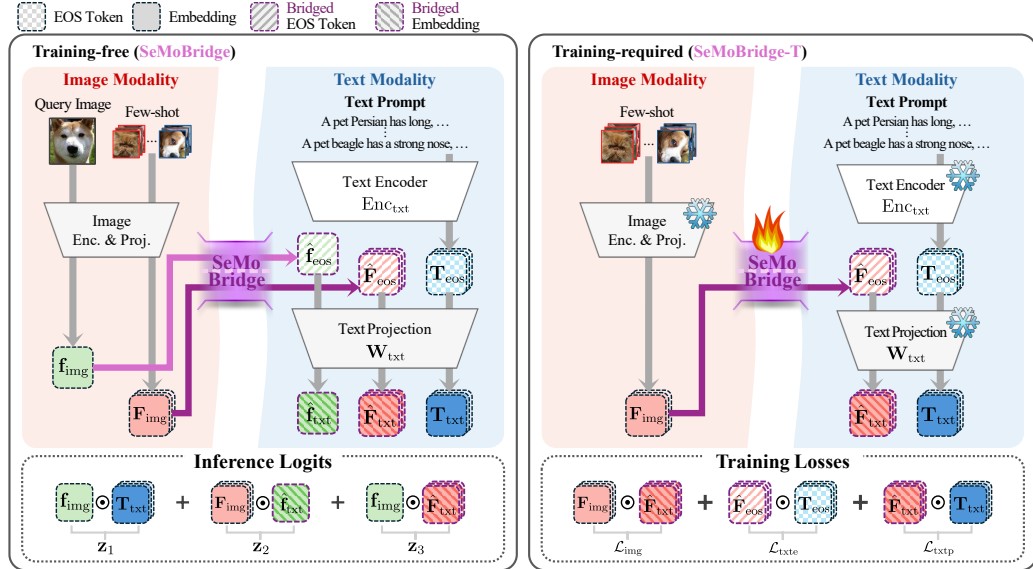

Figure 3: Overall architecture of our method. **Left:** At inference time, SeMoBridge maps both query and few-shot images into the text modality. The resulting pseudo-EOS tokens are passed through CLIP's text projection layer, enabling robust inter-modal comparisons. Classification is performed by blending three logits: CLIP's Zero-Shot Prior, Original Few-Shots vs. Bridged Query, and Original Query vs. Bridged Few-Shots. **Right:** SeMoBridge-T is supervised from both images and texts. Three primary loss terms are used: image alignment, encoded text alignment, and projected text alignment. Only the SeMoBridge parameters are updated, and all CLIP components remain frozen.

$$\hat{\mathbf{f}}_{\text{txt}} = \mathbf{W}_{\text{txt}}\hat{\mathbf{f}}_{\text{eos}} \approx \frac{\|\mathbf{T}_{\text{eos}}\|}{\|\mathbf{W}_{\text{txt}}^{+}\mathbf{f}_{\text{img}}\|}\mathbf{f}_{\text{img}} \qquad (3)$$

Here, by rearranging the scale factor to the front, we observe that the composition $\mathbf{W}_{\text{txt}}\mathbf{W}_{\text{txt}}^{+}$ approximately forms an identity matrix due to the properties of the pseudo-inverse. As a result, the transformation simplifies to a scaled version of the original image embedding, with its magnitude aligned to that of a text embedding. Through this process, we can now perform image-image comparisons inter-modally $\mathbf{f}_{\text{img}}\mathbf{F}_{\text{img}} \rightarrow \hat{\mathbf{f}}_{\text{txt}}\mathbf{F}_{\text{img}}$ (see Figure 2).

### 3.3 TRAINING-FREE SEMOBRIDGE INFERENCE.

SeMoBridge offers a powerful baseline that requires no training. It works by initializing the bridge with the pseudo-inverse of CLIP's text projection matrix.

To make a final prediction, we combine CLIP's zero-shot prior with two few-shot signals derived from our bridge. This is achieved by blending the logit scores (see Figure 3), each playing a specific role in refining the classification decision:

- $\mathbf{z}_1$ **Zero-Shot Prior.** Standard inter-modal CLIP zero-shot logit, calculated as the similarity between the original query image embedding ($\mathbf{f}_{\text{img}}$) and the class text prompts ($\mathbf{T}_{\text{txt}}$).

- $\mathbf{z}_2$ **Original Few-Shots ($\mathbf{F}_{\text{img}}$) vs. Bridged Query ($\hat{\mathbf{f}}_{\text{txt}}$).** This compares how well the bridged query image matches the class few-shot images through CLIP's inter-modal space.

- $\mathbf{z}_3$ **Original Query ($\mathbf{f}_{\text{img}}$) vs. Bridged Few-Shots ($\hat{\mathbf{F}}_{\text{txt}}$).** This offers a complementary signal by doing the reverse: we now compare the original query image against the bridged versions of the few-shot images, increasing robustness.

The final prediction is a weighted sum of these three logits: $\mathbf{z}_q = \lambda_1\mathbf{z}_1 + \lambda_2\mathbf{z}_2 + \lambda_3\mathbf{z}_3$.

Where $\lambda_i$ are scalar blending weights that balance the contribution of each similarity signal. Additionally, we adapt the same logit sharpening strategy as APE (Zhu et al., 2023). All parameters

are tuned via optimization on a validation set. This strategy enables SeMoBridge to robustly blend signals while dynamically adapting to class confidence.

### 3.4 Multi-modal supervised SeMoBridge-T training.

By using multi-modal supervision, SeMoBridge-T is trained to align the bridged embeddings with both their original images and class descriptions. This ensures robust semantic alignment with their respective class. To adapt the projection better for our task, we add a class-specific bias (CSB) term $\hat{\tau} \in \mathbb{R}^{C \times d_t}$ for each class after the transformation. This allows the bridge to capture nuanced semantic differences across a large number of classes (e.g. 1000 for ImageNet), overcoming the expressiveness bottleneck of a single projection.

Formally, during training, few-shot embeddings of a class $c$ are bridged into the text modality following the procedure described in Section 3.2, in addition to the CSB term:

$$\hat{\mathbf{F}}_{\text{eos}}^c \approx \frac{\|\mathbf{T}_{\text{eos}}\|}{\|\hat{\mathbf{W}}_{\text{txt}}^+ \mathbf{F}_{\text{img}}^c + \hat{\tau}^c\|} (\hat{\mathbf{W}}_{\text{txt}}^+ \mathbf{F}_{\text{img}}^c + \hat{\tau}^c) \tag{4}$$

Here, $\hat{\mathbf{W}}_{\text{txt}}^+$ and $\hat{\tau}$ are learnable and our parameters to optimize. $\hat{\mathbf{F}}_{\text{eos}}^c$ is projected to $\hat{\mathbf{F}}_{\text{txt}}^c$ through CLIP's text projection $\mathbf{W}_{\text{txt}}$. Notably, the CSB learned from the few-shot set during training is not applied to bridge the query image embedding $\mathbf{f}_{\text{img}}$, since its class is unknown.

We train SeMoBridge-T using the following multi-modal loss objective:

$$\mathcal{L} = \lambda_{\text{it}} \mathcal{L}_{\text{img}} + (1 - \lambda_{\text{it}})(\frac{\mathcal{L}_{\text{txte}} + \mathcal{L}_{\text{txtp}}}{2}) + \lambda_{\text{c}} \mathcal{L}_{\text{cons}} + \lambda_{\text{b}} \mathcal{L}_{\text{bias}} \tag{5}$$

First, $\mathcal{L}_{\text{img}}$ ensures that the bridged few-shots $\hat{\mathbf{F}}_{\text{txt}}$ retain semantic information of the few-shot embeddings $\mathbf{F}_{\text{img}}$, computed using the centroid of the $K$ shots per class. Second, $\mathcal{L}_{\text{txte}}$ encourages alignment to the class description EOS tokens $\mathbf{T}_{\text{eos}}$. $\mathcal{L}_{\text{txtp}}$ is the same, but in projected CLIP space. Together, these primary losses guide the bridge to learn representations that retain both visual and textual semantic information. Image and text influence is balanced by a hyperparameter $\lambda_{\text{it}} = \frac{1}{2}$, which we keep fixed for all datasets.

In addition, we include $\mathcal{L}_{\text{cons}}$ as a generalization that encourages all bridged few-shots $\hat{\mathbf{F}}_{\text{txt}} \in \mathbb{R}^{C \times K \times d}$ within the same class to be similar to each other. This promotes more robust representations for each class. The final term $\mathcal{L}_{\text{bias}}$ stabilizes training by regularizing the norms of the CSB vectors $\hat{\tau} \in \mathbb{R}^{C \times d_t}$, ensuring that they remain balanced across classes. This is particularly important because these biases are not applied when bridging query images during inference, and high variation could lead to instability of the bridge. Their respective coefficients $\lambda_{\text{c}} = \frac{1}{10}$ and $\lambda_{\text{b}} = \frac{1}{10}$ are both fixed as well.

## 4 Experiments

We evaluate SeMoBridge and SeMoBridge-T across 11 datasets commonly used in few-shot image classification (details in Appendix A.1). All experiments are done using CLIP's ViT-B/16 unless otherwise stated. Further implementation details are in Appendix A.2.

**Performance against state-of-the-art.** Figures 4 and 5 present accuracy across all datasets and shot counts for SeMoBridge and SeMoBridge-T, respectively. The training-free SeMoBridge outperforms APE on 7/11 datasets, with great improvements on low shot

Table 1: Comparison of training metrics. We report average accuracy (%) of all shot settings. Parameters are for 16-shot ImageNet on ViT-B/16.

| Method | Param. | Avg. Time | Avg. Acc. |
|---|---|---|---|
| CoOp | 0.01 M | 10 h 0 min | 63.90 |
| CLIP-Adapter | 0.52 M | 32 min | 69.45 |
| Tip-Adapter-F | 16.3 M | 4 min | 75.90 |
| LDC | 69 M | 2 min | 77.17 |
| APE-T | 0.51 M | 3 min 30 s | 77.18 |
| PromptSRC | 0.05 M | 1 h 42 min | 77.90 |
| **SeMoBridge-T w/o CSB** | 0.26 M | **22 s** | 78.14 |
| **SeMoBridge-T** | 0.77 M | 27 s | **78.15** |

counts (1, 2, and 4). Similarly, SeMoBridge-T overall outperforms all prior methods on low shot counts while requiring a fraction of the training time (see Figure 1). Results on RN-50 are reported in Appendices 11 and 12.

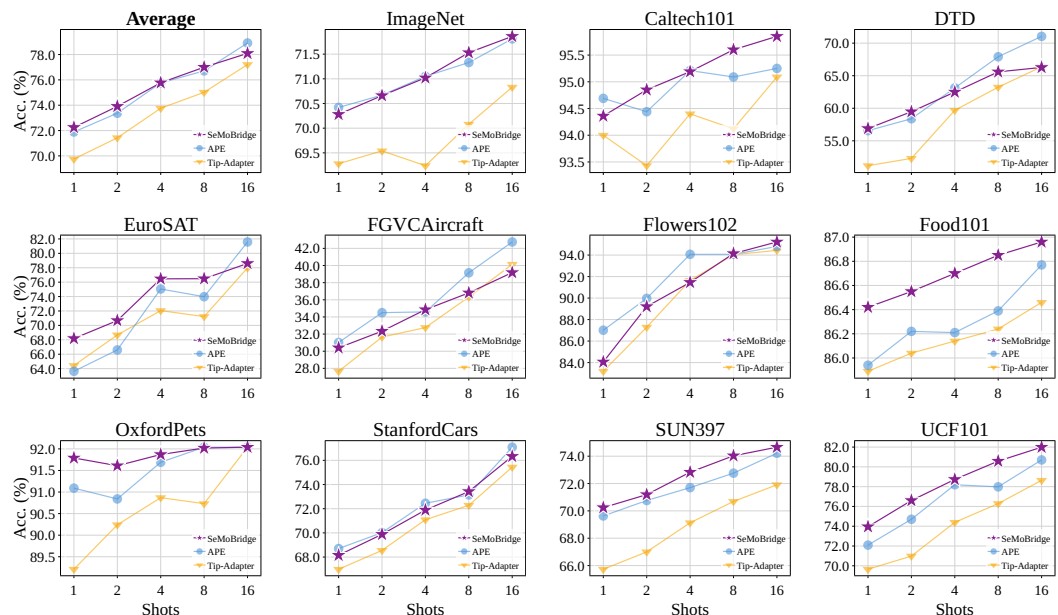

Figure 4: Few-shot accuracy of training-free SeMoBridge against other methods with ViT-B/16.

**Efficiency Analysis.** We report parameter count, training time, and accuracy in Table 1. SeMoBridge-T achieves superior accuracy to other methods, while requiring a fraction of the training time. This is the result of its lightweight architecture, backpropagating through only the text projection and small projection module. The minimal memory and compute footprint makes SeMoBridge highly practical for real-world applications.

**Robustness to Distribution Shift.** In Table 2, we evaluate robustness to distribution shift by using the 16-shot standard ImageNet few-shot set and testing on variants. Interestingly, SeMoBridge-T outperforms existing methods on both OOD sets by up to +0.71 % even though standard ImageNet accuracy is lower. This suggests that the bridged representations generalize well across domains and is robust.

Table 2: Comparison of accuracy (%) under 16-shot ImageNet out-of-distribution setting.

| Method | Source | Target | |
|---|---|---|---|
| | ImageNet | -V2 | -Sketch |
| *Zero-Shot* | | | |
| CLIP | 66.73 | 60.83 | 46.15 |
| *Training-free* | | | |
| APE | 71.81 | 64.81 | **49.95** |
| **SeMoBridge** | **71.86**±0.05 | **64.90**±0.08 | 49.55±0.02 |
| | +0.05 | +0.09 | -0.40 |
| *Training* | | | |
| CoOp | 71.51 | 64.20 | 47.99 |
| CoCoOp | 71.02 | 64.07 | 48.75 |
| MaPLe | 70.72 | 64.07 | 49.15 |
| LDC | 73.88 | 66.10 | 48.85 |
| APE-T | **74.13** | 66.21 | 49.73 |
| **SeMoBridge-T** | 73.98±0.05 | **66.49**±0.04 | **50.44**±0.14 |
| | -0.15 | +0.28 | +0.71 |

## 5 ABLATION STUDY

**The Role of Text Supervision.** SeMoBridge-T excels in low-data settings (1, 2, and 4 shots) due to its effective use of text supervision, an advantage that grows as the number of shots decreases. Figure 6 (right) reveals that descriptive, LLM-generated prompts, such as CuPL (Pratt et al., 2023), provide a greater performance benefit over simpler templates when fewer images are available. These rich prompts offer class-specific semantic information, such as attributes and context, which the model can leverage when visual data is scarce. SeMoBridge-T is designed to take advantage of this: during training, bridged embeddings are aligned with both image and text modalities. This allows the model to rely on strong semantic priors from text prompts when image supervision is weak. Ablation studies (Table 3) confirm this, showing that adding textual alignment losses ($\mathcal{L}_{\text{txte}}$ and $\mathcal{L}_{\text{txtp}}$) provides the largest accuracy boost in 1-shot scenarios. The benefit of text supervision diminishes in

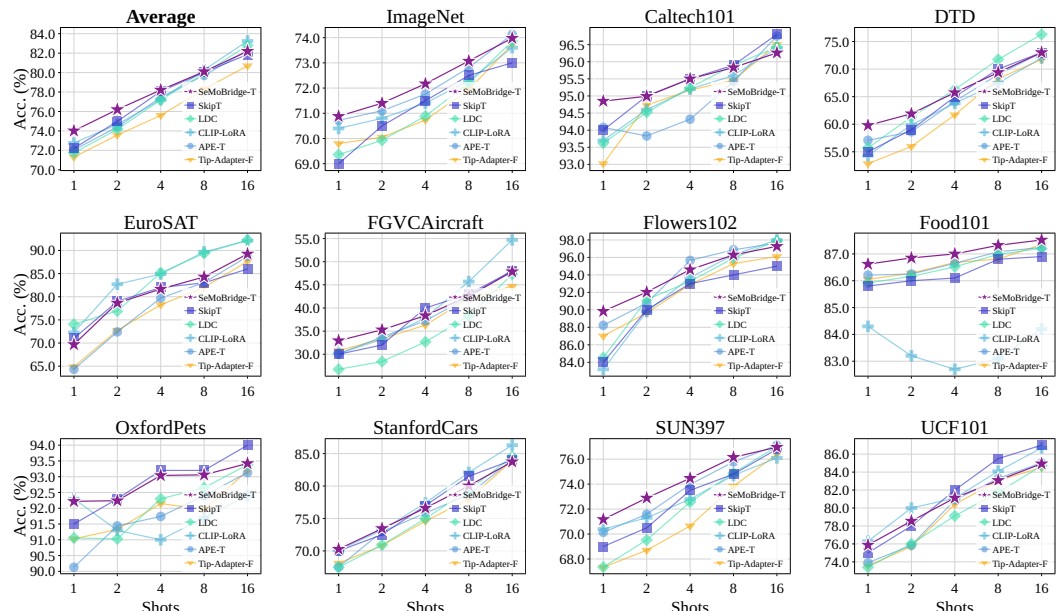

Figure 5: Few-shot accuracy of trained SeMoBridge-T against other methods with ViT-B/16.

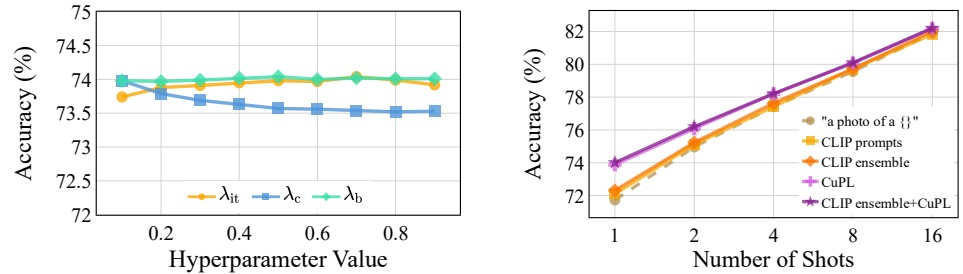

Figure 6: **Left:** Sensitivity analysis of $\lambda_{\text{it}}$, $\lambda_{\text{c}}$, and $\lambda_{\text{b}}$ on 16-shot ImageNet. Performance is stable across varying hyperparameters. **Right:** Analysis of different class text prompt templates on SeMoBridge-T's average accuracy over 11 datasets for different numbers of shots.

higher-shot settings (8-16 shots) as the model can increasingly rely on the visual information from the larger set of few-shot images.

**Cosine similarity distribution.** An analysis of cosine similarity distributions (Figure 7) shows the effectiveness of SeMoBridge in addressing intra-modal misalignment. Direct image-to-image comparisons (2) suffer from poor calibration, demonstrated by a large overlap in similarity scores between images of the same class (paired) and those from different classes (unpaired).

SeMoBridge resolves this by transforming image embeddings into the text modality, which preserves semantic information and achieves a much clearer separation between paired and unpaired samples (3), similarly to CLIP's pre-training (1).

The trained version, SeMoBridge-T, further enhances this effect, increasing the separation between the distributions (4) and confirming its ability to correct the misalignment and enable more reliable comparisons.

**Impact of loss terms.** Table 3 presents an ablation of the SeMoBridge-T training loss components. Image supervision ($\mathcal{L}_{\text{img}}$) is most critical when a large number of shots are available (16-shot). However, in very low-data settings (1-shot), the addition of text supervision ($\mathcal{L}_{\text{txte}}$, $\mathcal{L}_{\text{txtp}}$) becomes essential. It provides complementary semantic knowledge from LLMs, which improves performance when visual data is scarce. Combining both image and text supervision leads to consistent improvements across all settings.

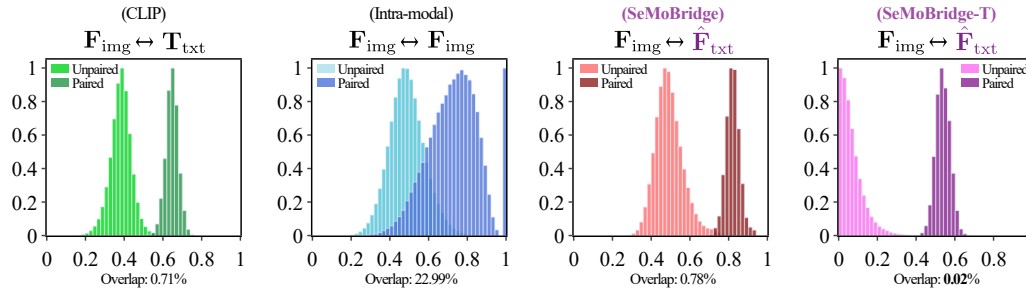

Figure 7: Histogram of cosine similarity distributions on ImageNet's few-shot set using different comparison methods. Each method shows the similarity for unpaired (different class) and paired (same class).

While the consistency loss $\mathcal{L}_{\text{cons}}$ shows no benefit in the 1-shot regime, where there is no intra-class variation in the few-shot set, it becomes important as the number of shots increases. In the 16-shot setting, it improves generalization by encouraging the bridged embeddings of all shots within the same class to stay similar. Finally, the bias norm regularization $\mathcal{L}_{\text{bias}}$ provides additional stability with the best-performing configuration.

**Ablation of Logits.** Table 4 presents SeMoBridge's accuracy when using only specific logit signals for prediction. The first row with only $z_1$ refers to zero-shot CLIP. $z_2$ and $z_3$ are SeMoBridge-derived logits. Notably, SeMoBridge and SeMoBridge-T are both able to achieve excellent accuracy even without CLIP's logit signal ($z_2 + z_3$). Although the 1-shot-scenario does not provide enough information for the training-free model in this case, SeMoBridge-T's training strategy yields large improvements.

In the trained model, bridging the few-shot images to the text modality ($z_3$) yields the best accuracy when using only a single logit signal. This is because SeMoBridge-T was trained to bridge the few-shot set into the text modality while preserving the semantic information. Bridging the unseen query image ($z_2$) is also effective for the 16-shot scenario.

Table 3: Ablation study of SeMoBridge-T's training loss terms and their impact on accuracy (%) over 11 datasets for 1 and 16 shot tasks.

| $\mathcal{L}_{\text{img}}$ | $\mathcal{L}_{\text{txtp}}$ | $\mathcal{L}_{\text{txte}}$ | $\mathcal{L}_{\text{cons}}$ | $\mathcal{L}_{\text{bias}}$ | 1 | 16 | avg. |
|---|---|---|---|---|---|---|---|
| | | | | | 72.25 | 78.09 | 75.17 |
| _Image_ | | | | | | | |
| ✓ | | | | | 72.74 | 81.79 | 77.27 |
| ✓ | | | ✓ | ✓ | 72.95 | **82.38** | 77.67 |
| _Text_ | | | | | | | |
| | ✓ | ✓ | | | 71.91 | 77.47 | 74.69 |
| | ✓ | ✓ | ✓ | ✓ | 72.17 | 77.15 | 74.66 |
| _Image + Text_ | | | | | | | |
| ✓ | ✓ | ✓ | | | 73.96 | 81.88 | 77.92 |
| ✓ | ✓ | ✓ | ✓ | | 73.99 | 82.18 | 78.09 |
| ✓ | ✓ | ✓ | ✓ | ✓ | **74.01** | 82.20 | **78.11** |

Table 4: Ablation of the impact of logit signals on accuracy over 11 datasets for 1 and 16 shot tasks. Results are shown for both the training-free _SeMoBridge_ and the trained _SeMoBridge-T_.

| $z_1$ | $z_2$ | $z_3$ | SeMoBridge 1 | SeMoBridge 16 | SeMoBridge avg. | SeMoBridge-T 1 | SeMoBridge-T 16 | SeMoBridge-T avg. |
|---|---|---|---|---|---|---|---|---|
| ✓ | | | 65.52 | 65.52 | 65.52 | 65.52 | 65.52 | 65.52 |
| | ✓ | | 40.92 | 70.78 | 55.85 | 42.26 | 77.12 | 59.69 |
| | | ✓ | 37.23 | 62.59 | 49.91 | 72.36 | 81.31 | 76.84 |
| | ✓ | ✓ | 40.97 | 70.72 | 55.85 | 72.58 | 82.05 | 77.32 |
| ✓ | ✓ | | 72.25 | 78.06 | 75.16 | 73.65 | 81.62 | 77.64 |
| ✓ | ✓ | ✓ | **72.25** | **78.09** | **75.17** | **74.02** | **82.35** | **78.19** |

## 6 CONCLUSION

We propose SeMoBridge, a _Semantic Modality Bridge_ that efficiently adapts CLIP for few-shot classification by resolving intra-modal misalignment. By bridging image embeddings into the text modality via a closed-form transformation, SeMoBridge enables more accurate few-shot learning by leveraging CLIP's strong inter-modal alignment. Its lightweight trainable variant, SeMoBridge-T, uses multi-modal supervision to further enhance performance. Extensive experiments across 11 datasets confirm that our method achieves state-of-the-art results with minimal computational cost, outperforming existing baselines. Future work will extend SeMoBridge to other CLIP-based tasks like multi-modal retrieval and object detection.

**Reproducibility Statement.** To ensure the reproducibility of our results and ensure a fair comparison with prior work, all of our experiments are built upon the Dassl framework by the CoOp authors (Zhou et al., 2022). Given that few-shot accuracy is highly sensitive to the specific samples available, its use guarantees that we evaluate our approach on the exact same few-shot data splits as methods like CoOp and Tip Adapter (Zhang et al., 2021). Comprehensive details regarding the datasets, data augmentation strategies, hyperparameters, and other implementation specifics are documented in the Appendices A.1 and A.2. Our full source code with running instructions is included in the supplementary material.

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

# A APPENDIX

## A.1 DATASET DETAILS

We evaluate SeMoBridge and SeMoBridge-T across 11 datasets commonly used in few-shot image classification: OxfordPets (Parkhi et al., 2012), OxfordFlowers (Nilsback & Zisserman, 2008), FGVCAircraft (Maji et al., 2013), DTD (Cimpoi et al., 2014), EuroSAT (Helber et al., 2019), StanfordCars (Krause et al., 2013), Food101 (Bossard et al., 2014), SUN397 (Xiao et al., 2010), Caltech101 (Fei-Fei et al., 2004), UCF101 (Soomro et al., 2012), and ImageNet (Deng et al., 2009). For robustness evaluation, we follow standard practice and test on out-of-distribution (OOD) splits -V2 and -Sketch (Recht et al., 2019) derived from ImageNet. In all experiments, we follow the few-shot setup of CoOp (Zhou et al., 2022), using 1, 2, 4, 8, or 16 labeled image samples per class. For each dataset, shot count, and vision encoder, we run three experiments with seeds 1, 2, and 3. We report the standard deviation of the accuracy based on them.

In Table 5, we present dataset sizes, the calculated $\|\mathbf{T}^{\text{eos}}\|$ from Equation 4, and which data augmentation is applied to the few-shot sets. Augmented shots are treated the same as "real" shots, essentially increasing the size of $K$ by creating altered images.

Table 5: Dataset statistics including average CLIP text token length $\|\mathbf{T}^{\text{eos}}\|$ and data augmentation strategy.

| Dataset | Classes | Train | Test | $\|\mathbf{T}^{\text{eos}}\|$ | | Few-shot Augmentation | | | | |
| --- | --- | --- | --- | --- | --- | --- | --- | --- | --- | --- |
| | | | | ViT-B/16 | RN-50 | Aug. Epochs | Hor. Flip | Rand. Res. Crop | Rand. Hor. Flip | Col. Jitter |
| ImageNet | 1,000 | 1.28M | 50,000 | 19.82 | 18.78 | 1 | ✓ | | | |
| Caltech101 | 100 | 4,128 | 2,465 | 19.37 | 18.40 | 0 | | | | |
| DTD | 47 | 2,820 | 1,692 | 20.10 | 18.89 | 10 | | ✓ | ✓ | ✓ |
| EuroSAT | 10 | 1,600 | 8,100 | 20.26 | 19.08 | 1 | ✓ | | | |
| FGVCAircraft | 100 | 3,334 | 3,333 | 20.42 | 19.31 | 1 | ✓ | | | |
| Flowers102 | 102 | 4,093 | 2,463 | 21.02 | 19.59 | 10 | | ✓ | ✓ | ✓ |
| Food101 | 101 | 50,500 | 30,300 | 19.89 | 18.87 | 0 | | | | |
| OxfordPets | 37 | 2,944 | 3,669 | 20.79 | 19.53 | 0 | | | | |
| StanfordCars | 196 | 6,509 | 8,041 | 20.67 | 19.55 | 1 | ✓ | | | |
| SUN397 | 397 | 15,880 | 19,850 | 19.49 | 18.56 | 1 | ✓ | | | |
| UCF101 | 101 | 7,639 | 3,783 | 19.99 | 18.86 | 1 | ✓ | | | |

## A.2 IMPLEMENTATION DETAILS

For class text descriptions, we use a combination of CLIP's handmade templates and CuPL-generated LLM prompts (Pratt et al., 2023), following APE (Zhu et al., 2023). We apply horizontal flipping, random resized crop, and color jittering as data augmentation to the few-shot set. This is applied depending on the dataset characteristics.

All input images are resized such that the longer side is 224 pixels, followed by center-crop to $224 \times 224$, and normalization following CLIP preprocessing. SeMoBridge-T is trained using AdamW for 5000 epochs, with a fixed learning rate of 0.15e−4 and linear warmup for the first 500 epochs. We preload all few-shot samples into GPU memory, eliminating the need for batching. After training, we first select the best model epoch based on the accuracy on the held-out validation set, and then optimize the logit blending parameters on it.

We run the experiments using an RTX 4090 (24GB) and a Ryzen 5 5600X with 32GB RAM on Ubuntu 22.04 LTS. GPU memory used during training is 10GB. Main packages are Python 3.12.8 and PyTorch version 2.7.0 on CUDA 12.8.

## A.3 THE ROLE OF CLASS-SPECIFIC BIAS AND $\mathcal{L}_{\text{bias}}$

To better understand the behaviour of the class-specific bias (CSB) vectors used in SeMoBridge-T, we analyze their $\ell_2$-norms across the classes of the FGVCAircraft dataset. We compare 16-shot models trained with and without the regularization term $\mathcal{L}_{\text{bias}}$.

As shown in Figure 8, the regularized biases (green) have no variance. The class-specific vectors are uniformly scaled, which helps the bridge to stay balanced across classes. In contrast, the unregularized norms (red) vary much more, indicating that some classes dominate the bridge more than others.

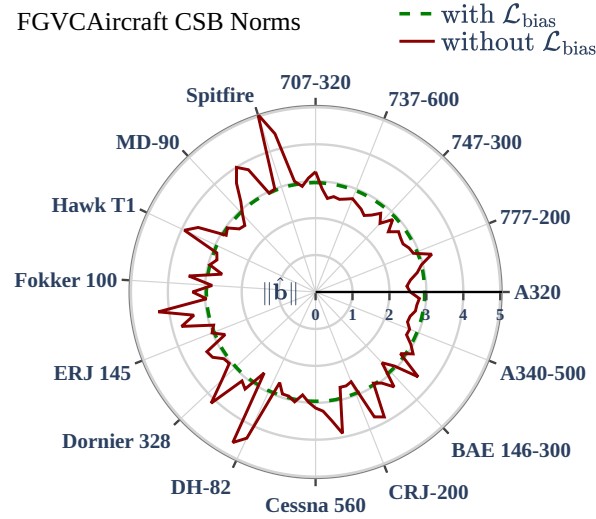

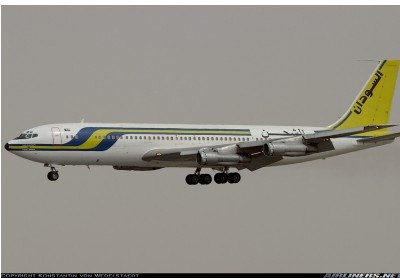

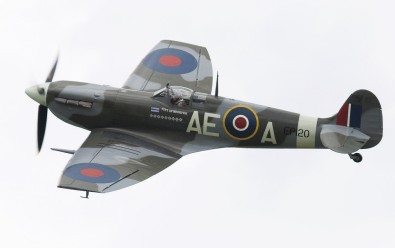

Figure 9: Examples from FGVCAircraft.
**Top:** *707-320* (visually regular).
**Bottom:** *Spitfire* (visually distinct).

Figure 8: Class-specific bias norm $\|\hat{\tau}\| \in \mathbb{R}^C$ comparison with and without $\mathcal{L}_{\text{bias}}$ on FGVCAircraft's 100 classes.

This is a problem during inference. Since the class of the query image is unknown, we cannot apply the class-specific bias to it. The bridge must operate in a way that is semantically centered across all classes. If the learned biases are highly unbalanced, the bridged query embedding may be pulled towards a subset of the classes, hindering generalization.

Interestingly, the bias norm is smaller for "regular looking" or common aircraft such as the *707-320*, *CRJ-200*, and *MD-90*. For more visually distinct aircraft like the *Hawk T1* and *Spitfire*, the bias norm is much larger. This suggests that the unregularized bridge is centered around the more typical aircraft, which makes the bridging less effective for unusual classes. A *Spitfire* query, for example, may be poorly aligned if the bridge has shifted away from that region of the space.

Regularizing the bias norms encourages the model to keep all classes equally represented in the bridging space. This helps maintain alignment even for visually unique classes, improving generalization at inference time.

We report class-specific bias norms for all 11 datasets in Figure 10.

Table 6: Impact of CSB across all datasets.

| Method | Shots | Pets | Flowers | Aircraft | DTD | EuroSAT | Cars | Food101 | SUN397 | Caltech | UCF101 | ImageNet | Avg |
|---|---|---|---|---|---|---|---|---|---|---|---|---|---|
| CLIP zero-shot | 0 | 89.10±0.00 | 70.73±0.00 | 24.69±0.00 | 44.09±0.00 | 48.31±0.00 | 65.61±0.00 | 85.87±0.00 | 62.59±0.00 | 93.35±0.00 | 67.62±0.00 | 68.73±0.00 | 65.52 |
| SeMoBridge-T w/o CSB | 1 | 92.30±0.13 | 89.61±0.81 | 32.81±0.74 | 59.91±0.45 | 69.78±5.48 | 69.78±0.34 | 86.67±0.02 | 71.01±0.14 | 94.70±0.25 | 76.04±0.84 | 70.67±0.03 | 73.93 |
| SeMoBridge-T | 1 | 92.22±0.19 | 89.84±0.85 | 32.94±0.43 | 59.79±0.64 | 69.69±5.48 | 70.27±0.44 | 86.62±0.04 | 71.17±0.21 | 94.85±0.20 | 75.87±0.56 | 70.88±0.09 | 74.01 |
| SeMoBridge-T w/o CSB | 2 | 92.22±0.22 | 92.18±0.74 | 35.42±0.48 | 61.82±1.47 | 78.69±2.85 | 73.66±0.23 | 86.85±0.06 | 72.36±0.12 | 94.74±0.43 | 78.25±0.91 | 71.28±0.12 | 76.13 |
| SeMoBridge-T | 2 | 92.24±0.22 | 92.03±0.65 | 35.28±0.71 | 61.90±1.09 | 78.65±2.96 | 73.46±0.75 | 86.85±0.09 | 72.89±0.20 | 94.99±0.42 | 78.55±0.77 | 71.40±0.04 | 76.20 |
| SeMoBridge-T w/o CSB | 4 | 92.57±0.19 | 94.70±0.28 | 38.80±0.29 | 65.70±0.95 | 81.80±1.34 | 77.02±0.52 | 87.00±0.04 | 74.27±0.28 | 95.42±0.09 | 81.10±0.42 | 72.04±0.04 | 78.22 |
| SeMoBridge-T | 4 | 93.04±0.28 | 94.60±0.25 | 38.35±0.48 | 65.74±0.97 | 81.66±1.00 | 76.61±0.32 | 87.00±0.07 | 74.47±0.23 | 95.50±0.17 | 81.12±0.31 | 72.17±0.07 | 78.21 |
| SeMoBridge-T w/o CSB | 8 | 92.92±0.39 | 96.25±0.38 | 43.62±0.80 | 69.27±0.17 | 84.35±1.07 | 80.31±0.67 | 87.39±0.16 | 75.87±0.07 | 95.71±0.04 | 83.36±0.52 | 73.11±0.10 | 80.20 |
| SeMoBridge-T | 8 | 93.06±0.33 | 96.29±0.24 | 42.60±0.59 | 69.40±0.18 | 84.29±1.16 | 80.03±0.59 | 87.32±0.18 | 76.15±0.18 | 95.83±0.29 | 83.08±0.70 | 73.07±0.08 | 80.10 |
| SeMoBridge-T w/o CSB | 16 | 93.58±0.16 | 96.94±0.14 | 48.61±0.54 | 72.78±0.56 | 89.37±0.36 | 83.85±0.45 | 87.58±0.07 | 77.14±0.06 | 96.31±0.09 | 85.07±0.13 | 73.96±0.22 | 82.29 |
| SeMoBridge-T | 16 | 93.42±0.44 | 97.27±0.45 | 47.84±0.63 | 73.01±0.15 | 89.25±0.25 | 83.75±0.33 | 87.52±0.08 | 76.96±0.12 | 96.26±0.09 | 84.93±0.35 | 73.98±0.05 | 82.20 |

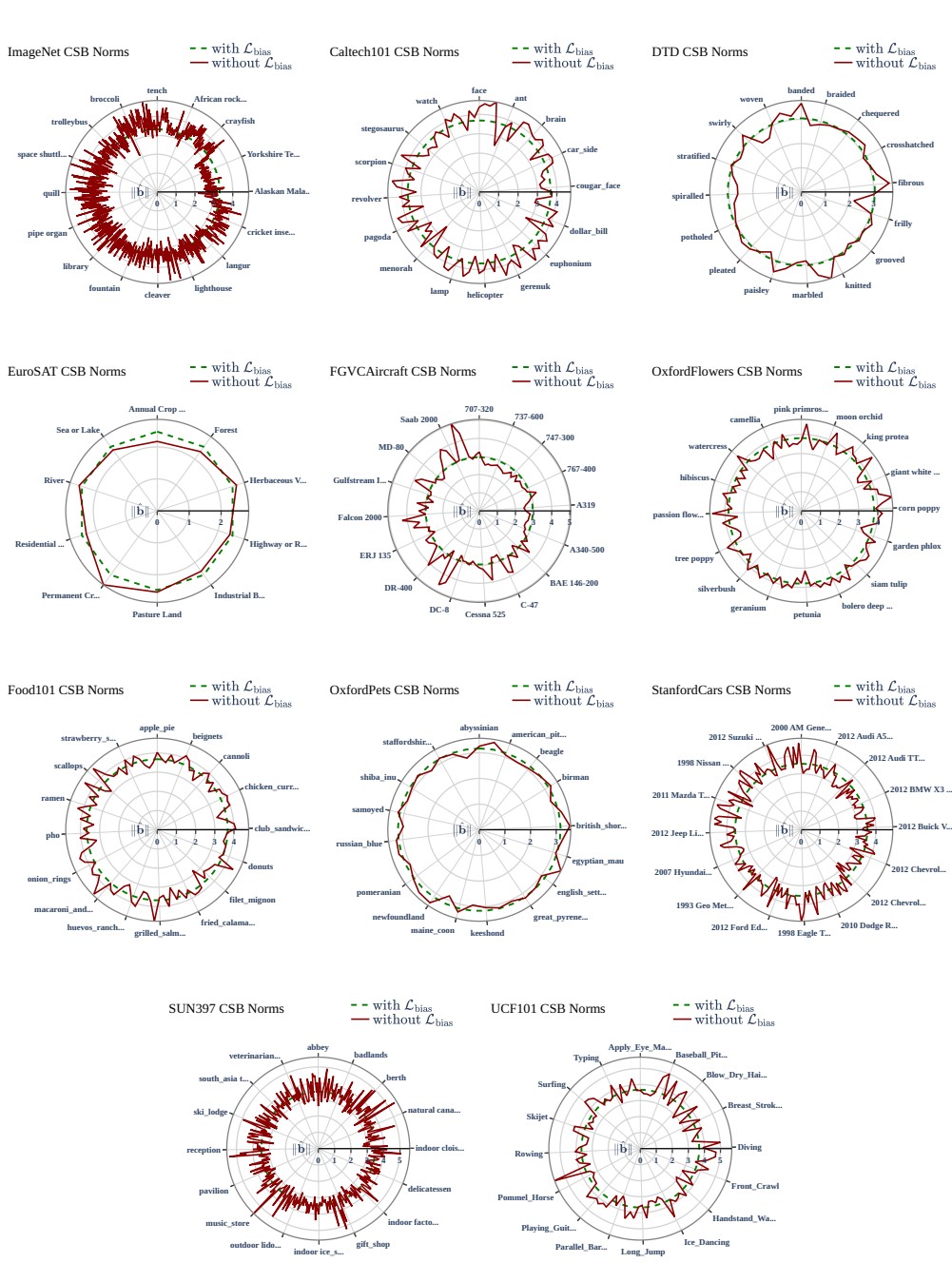

Figure 10: Class-specific bias norm $\|\hat{\mathbf{f}}\| \in \mathbb{R}^C$ comparison with and without $\mathcal{L}_{\text{bias}}$ on all 16-shot datasets.

## A.4 GPT-3 PROMPTS USED IN CUPL

In Table 7, we show all prompts used for GPT-3 to generate the class descriptions for each dataset.

Table 7: GPT-3 Commands Used in CuPL.

| Dataset | GPT-3 Commands |
|---|---|
| ImageNet | "Describe what a {} looks like"
"How can you identify {}?"
"What does {} look like?"
"Describe an image from the internet of a {}"
"A caption of an image of {}:" |
| Caltech101 | "Describe what a {} looks like"
"What does a {} look like"
"Describe a photo of a {}" |
| DTD | "What does a {} material look like?"
"What does a {} surface look like?"
"What does a {} texture look like?"
"What does a {} object look like?"
"What does a {} thing look like?"
"What does a {} pattern look like?" |
| EuroSAT | "Describe an aerial satellite view of {}"
"How does a satellite photo of a {} look like"
"Visually describe a satellite view of a {}" |
| FGVCAircraft | "Describe a {} aircraft" |
| Flowers102 | "What does a {} flower look like"
"Describe the appearance of a {}"
"A caption of an image of {}"
"Visually describe a {}, a type of flower" |
| Food101 | "Describe what a {} looks like"
"Visually describe a {}"
"How can you tell the food in the photo is a {}?" |
| OxfordPets | "Describe what a {} pet looks like"
"Visually describe a {}, a type of pet" |
| StanfordCars | "How can you identify a {}"
"Description of a {}, a type of car"
"A caption of a photo of a {}:"
"What are the primary characteristics of a {}?"
"Description of the exterior of a {}"
"What are the characteristics of a {}, a car?"
"Describe an image from the internet of a {}"
"What does a {} look like?"
"Describe what a {}, a type of car, looks like" |
| SUN397 | "Describe what a {} looks like"
"How can you identify a {}?"
"Describe a photo of a {}" |
| UCF101 | "What does a person doing {} look like"
"Describe the process of {}"
"How does a person {}" |
| ImageNet-V2 | "Describe what a {} looks like"
"How can you identify {}?"
"What does {} look like?"
"Describe an image from the internet of a {}"
"A caption of an image of {}:" |
| ImageNet-Sketch | "Describe what a {} looks like"
"How can you identify {}?"
"What does {} look like?"
"Describe an image from the internet of a {}"
"A caption of an image of {}:" |

## A.5 Full algorithms for inference and training.

In Algorithms 1 and 2, we describe our inference and training processes in detail.

---

**Algorithm 1** Training-free SeMoBridge Inference

---

1: **Definition.**
   Pretrained CLIP encoders: $\text{Enc}_{\text{img}}$, $\text{Enc}_{\text{txt}}$,
   Pretrained projection matrices: $\mathbf{W}_{\text{img}}$, $\mathbf{W}_{\text{txt}}$,
   Pseudo-inverse projection: $\mathbf{W}_{\text{txt}}^{+} \leftarrow \text{pinv}(\mathbf{W}_{\text{txt}})$,
   Sharpening function: $\phi(\mathbf{z}, \lambda) = \exp(-\lambda(1 - \mathbf{z}))$,
2: **Input:**
   Query image $x_q$,
   Few-shot set $\mathcal{D} = \{(x_i, y_i)\}_{i=1}^{C \times K}$,
   Text prompts $\{t_c\}_{c=1}^{C}$,
   Class-wise one-hot labels $\mathbf{L} \in \mathbb{R}^{C \times C}$
3: **Output:** Prediction logits $\mathbf{z}_q \in \mathbb{R}^C$
4: Encode and project the query&few-shot set:
   $\mathbf{f}_{\text{img}}^{q} \in \mathbb{R}^{d} \leftarrow \mathbf{W}_{\text{img}}(\text{Enc}_{\text{img}}(x_q))$
   $\mathbf{F}_{\text{img}} \in \mathbb{R}^{C \times K \times d} \leftarrow \{\mathbf{W}_{\text{img}}(\text{Enc}_{\text{img}}(x_i))\}_{i=1}^{C \times K}$
5: Encode and project the text prompts:
   $\mathbf{T}_{\text{eos}} \in \mathbb{R}^{C \times d_t} \leftarrow \{\text{EOS}(\text{Enc}_{\text{txt}}(t_c))\}_{c=1}^{C}$
   $\mathbf{T}_{\text{txt}} \in \mathbb{R}^{C \times d} \leftarrow \{\mathbf{W}_{\text{txt}}(\mathbf{T}_{\text{eos}}^{c})\}_{c=1}^{C}$
6: Compute text token norm estimate:
   $\|\mathbf{T}_{\text{eos}}\| \leftarrow \frac{1}{C} \sum_{i=1}^{C} \|\mathbf{T}_{\text{eos}}^{i}\|$
7: Compute bridged query image:
   $\hat{\mathbf{f}}_{\text{eos}}^{q} \in \mathbb{R}^{d_t} \leftarrow \frac{\|\mathbf{T}_{\text{eos}}\|}{\|\mathbf{W}_{\text{txt}}^{+} \mathbf{f}_{\text{img}}^{q}\|} \cdot \mathbf{W}_{\text{txt}}^{+} \mathbf{f}_{\text{img}}^{q}$

   $\hat{\mathbf{f}}_{\text{txt}}^{q} \in \mathbb{R}^{d} \leftarrow \mathbf{W}_{\text{txt}}(\hat{\mathbf{f}}_{\text{eos}}^{q})$
8: **for all** few-shot embeddings $\mathbf{F}_{\text{img}}^{i} \in \mathbf{F}_{\text{txt}}$ **do**
9: $\quad \hat{\mathbf{F}}_{\text{eos}}^{i} \in \mathbb{R}^{d_t} \leftarrow \frac{\|\mathbf{T}_{\text{eos}}\|}{\|\mathbf{W}_{\text{txt}}^{+} \mathbf{F}_{\text{img}}^{i}\|} \cdot \mathbf{W}_{\text{txt}}^{+} \mathbf{F}_{\text{img}}^{i}$
10: $\quad \hat{\mathbf{F}}_{\text{txt}}^{i} \in \mathbb{R}^{d} \leftarrow \mathbf{W}_{\text{txt}}(\hat{\mathbf{F}}_{\text{eos}}^{i})$
11: **end for**
12: Compute class-wise mean of few-shot embeds:
   $\mathbf{F}'_{\text{img}} \in \mathbb{R}^{C \times d} \leftarrow \text{Classwisemean}(\mathbf{F}_{\text{img}})$
   $\hat{\mathbf{F}}'_{\text{txt}} \in \mathbb{R}^{C \times d} \leftarrow \text{Classwisemean}(\hat{\mathbf{F}}_{\text{txt}})$
13: Normalize:
   $\mathbf{F}'_{\text{img}} \leftarrow \text{Normalize}(\cdot)$
   $\hat{\mathbf{F}}'_{\text{txt}} \leftarrow \text{Normalize}(\cdot)$
   $\mathbf{T}_{\text{txt}} \leftarrow \text{Normalize}(\cdot)$
14: Optimize logit blending parameters on validation set:
   $\alpha, \beta, \gamma, \delta, \lambda_1, \lambda_2, \lambda_3, \lambda_4$
15: Compute soft label matrix:
   $\tilde{\mathbf{L}} \in \mathbb{R}^{C \times C} = \exp\left(\theta \cdot D_{\text{KL}}(\mathbf{F}'_{\text{img}} \mathbf{T}_{\text{txt}}^{\top} \| \mathbf{L})\right)$
16: Compute logits:
   $\mathbf{z}_1 \leftarrow \phi(\mathbf{f}_{\text{img}}^{q} \mathbf{T}_{\text{txt}}^{\top}, \alpha)$
   $\mathbf{z}_2 \leftarrow \phi(\hat{\mathbf{f}}_{\text{txt}}^{q} \mathbf{F}'_{\text{img}}^{\top}, \gamma) \cdot \tilde{\mathbf{L}}$
   $\mathbf{z}_3 \leftarrow \phi(\mathbf{f}_{\text{img}}^{q} \hat{\mathbf{F}}'_{\text{txt}}^{\top}, \beta) \cdot \tilde{\mathbf{L}}$
17: Compute final logits:
   $\mathbf{z}_q \leftarrow \lambda_1 \mathbf{z}_1 + \lambda_2 \mathbf{z}_2 + \lambda_3 \mathbf{z}_3$
18: **return** $\mathbf{z}_q$

---

---

**Algorithm 2** Training Procedure for SeMoBridge-T

---

1: **Definition.**
  Pretrained CLIP encoders: $\text{Enc}_{\text{img}}, \text{Enc}_{\text{txt}}$,
  Projection matrices: $\mathbf{W}_{\text{img}}, \mathbf{W}_{\text{txt}} \in \mathbb{R}^{d_t \times d}$,
  Pseudo-inverse projection: $\mathbf{W}_{\text{txt}}^+ \leftarrow \text{pinv}(\mathbf{W}_{\text{txt}})$,
  Trainable inverse projection: $\hat{\mathbf{W}}_{\text{txt}}^+ \leftarrow \mathbf{W}_{\text{txt}}^+$,
2: **Input:**
  Few-shot set $\mathcal{D} = \{(x_i, y_i)\}_{i=1}^{C \times K}$,
  Text prompts $\{t_c\}_{c=1}^C$,
  Class-wise one-hot labels $\mathbf{L} \in \mathbb{R}^{C \times C}$
  Consistency loss target $\mathbf{L}_{\text{cons}} \in \mathbb{R}^{CK \times C}$
3: **Output:**
  Trained inverse projection $\hat{\mathbf{W}}_{\text{txt}}^+ \in \mathbb{R}^{d \times d_t}$
  Trained class-specific bias $\hat{\mathbf{f}}_c \in \mathbb{R}^{C \times d_t}$
4: Encode and project the few-shot set:
  $\mathbf{F}_{\text{img}} \in \mathbb{R}^{C \times K \times d} \leftarrow \{\mathbf{W}_{\text{img}}(\text{Enc}_{\text{img}}(x_i))\}_{i=1}^{C \times K}$
5: Encode and project the text prompts:
  $\mathbf{T}_{\text{eos}} \in \mathbb{R}^{C \times d_t} \leftarrow \{\text{EOS}(\text{Enc}_{\text{txt}}(t_c))\}_{c=1}^C$
  $\mathbf{T}_{\text{txt}} \in \mathbb{R}^{C \times d} \leftarrow \{\mathbf{W}_{\text{txt}}(\mathbf{T}_{\text{eos}}^c)\}_{c=1}^C$
6: Compute norm estimate: $\|\mathbf{T}_{\text{eos}}\| \leftarrow \frac{1}{C} \sum_{i=1}^C \|\mathbf{T}_{\text{eos}}^i\|$
7: **for** each training epoch **do**
8:   Compute bridged few-shot embeddings:
    $\hat{\mathbf{F}}_{\text{eos}}^{c,k} \leftarrow \frac{\|\mathbf{T}_{\text{eos}}\|}{\|\hat{\mathbf{W}}_{\text{txt}}^+ \mathbf{F}_{\text{img}}^{c,k}\|} \cdot \hat{\mathbf{W}}_{\text{txt}}^+ \mathbf{F}_{\text{img}}^{c,k} + \hat{\mathbf{f}}_c$
    $\hat{\mathbf{F}}_{\text{txt}}^{c,k} \leftarrow \mathbf{W}_{\text{txt}}(\hat{\mathbf{F}}_{\text{eos}}^{c,k})$
9:   Compute class-wise mean embeddings:
    $\mathbf{F'}_{\text{img}} \in \mathbb{R}^{C \times d} \leftarrow \text{Classwisemean}(\mathbf{F}_{\text{img}})$
    $\hat{\mathbf{F}}'_{\text{txt}} \in \mathbb{R}^{C \times d} \leftarrow \text{Classwisemean}(\hat{\mathbf{F}}_{\text{txt}})$
    $\hat{\mathbf{F}}'_{\text{eos}} \in \mathbb{R}^{C \times d_t} \leftarrow \text{Classwisemean}(\hat{\mathbf{F}}_{\text{eos}})$
10:  Normalize:
    $\mathbf{F'}_{\text{img}} \leftarrow \text{Normalize}(\cdot)$
    $\hat{\mathbf{F}}'_{\text{txt}}, \hat{\mathbf{F}}'_{\text{eos}} \leftarrow \text{Normalize}(\cdot)$
    $\mathbf{T'}_{\text{txt}}, \mathbf{T'}_{\text{eos}} \leftarrow \text{Normalize}(\cdot)$
11:  Compute loss terms:
    **Image loss:**
      $\mathcal{L}_{\text{img}} \leftarrow \text{CE}\left(\mathbf{F'}_{\text{img}}^c \cdot \hat{\mathbf{F}}'^{c\top}_{\text{txt}}, \mathbf{L}_c\right)$
    **Encoded text loss:**
      $\mathcal{L}_{\text{txte}} \leftarrow \text{CE}\left(\hat{\mathbf{B}}'^c_{\text{eos}} \cdot \mathbf{T'}^{c\top}_{\text{eos}}, \mathbf{L}_c\right)$
    **Projected text loss:**
      $\mathcal{L}_{\text{txtp}} \leftarrow \text{CE}\left(\hat{\mathbf{F}}'^c_{\text{txt}} \cdot \mathbf{T'}^{c\top}_{\text{txt}}, \mathbf{L}_c\right)$
    **Consistency loss:**
      $\mathcal{L}_{\text{cons}} \leftarrow \text{CE}\left(\hat{\mathbf{f}}_{\text{txt}}^c \cdot \mathbf{F'}^{c\top}_{\text{img}}, \mathbf{L}_{\text{cons}}\right)$
    **Bias regularization:**
      Compute mean norm: $\bar{\tau} \leftarrow \frac{1}{C} \sum_{c=1}^C \|\hat{\tau}_c\|$
      $\mathcal{L}_{\text{bias}} \leftarrow \frac{1}{C} \sum_{c=1}^C (\|\hat{\tau}_c\| - \bar{\tau})^2$
12:  Compute total loss:
    $\mathcal{L} \leftarrow \lambda_{\text{it}} \mathcal{L}_{\text{img}} + (1 - \lambda_{\text{it}}) \cdot \frac{\mathcal{L}_{\text{txte}} + \mathcal{L}_{\text{txtp}}}{2}$
    $\quad + \lambda_{\text{c}} \cdot \mathcal{L}_{\text{cons}} + \lambda_{\text{b}} \cdot \mathcal{L}_{\text{bias}}$
13:  Update $\hat{\mathbf{W}}_{\text{txt}}^+, \hat{\tau}_c$ via gradient descent
14: **end for**
15: **return** $\hat{\mathbf{W}}_{\text{txt}}^+, \hat{\tau}_c$

---

## A.6 FEW-SHOT RESULTS USING RESNET-50.

In Figures 11 and 12, we plot RN-50 results for all datasets in comparison with other few-shot methods.

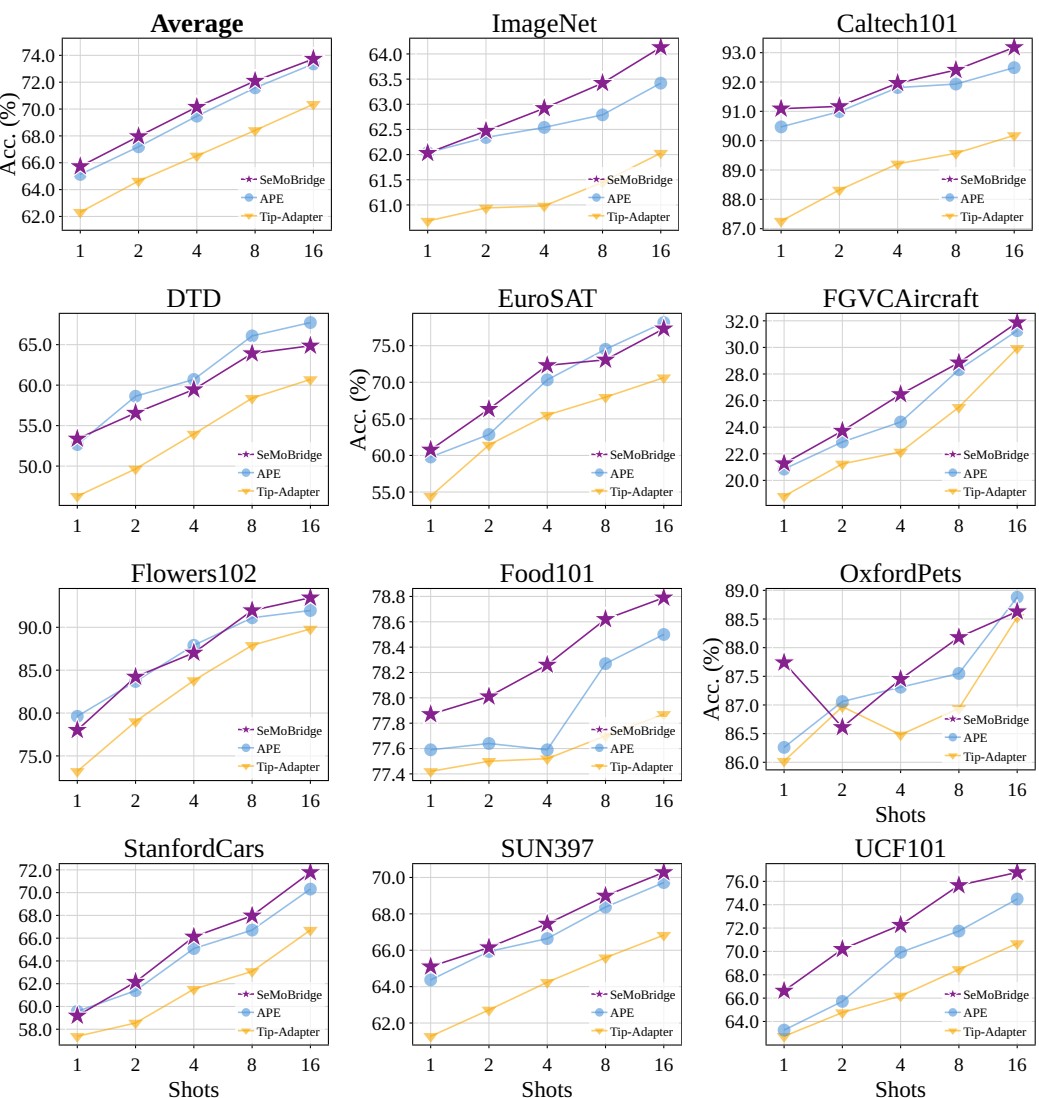

Figure 11: Few-shot accuracy of SeMoBridge against other training-free methods with RN-50.

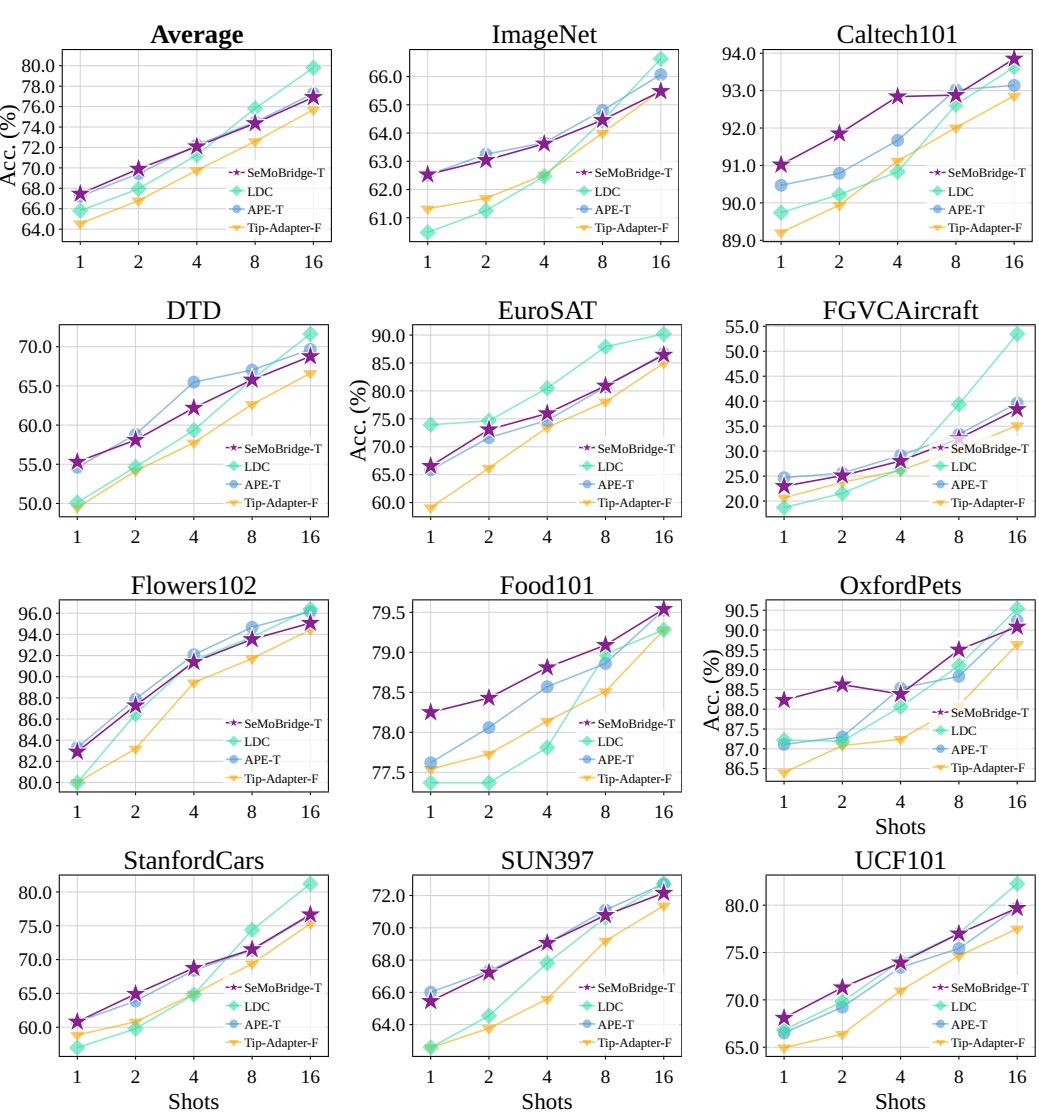

Figure 12: Few-shot accuracy of SeMoBridge-T against other trained methods with RN-50.

## A.7 RETRIEVAL EXPERIMENTS

To validate the versatility of SeMoBridge, we conducte additional experiments on Retrieval (both Image-Image and Text-Text), following the standard evaluation setting used in Cross the Gap (Mistretta et al., 2025). The objective for these tasks is to retrieve the top-$k$ items from a gallery that are most semantically similar to a given query.

### A.7.1 IMAGE-RETRIEVAL

In this setting, we aim to retrieve relevant images from a gallery given an image query. Standard CLIP-based retrieval typically relies on intra-modal comparison (Image-Image), which suffers from the misalignment issues discussed in the main text.

We apply SeMoBridge to project the query image into the text modality. This transforms the task into an inter-modal comparison between the bridged query (now in text space) and the gallery images (in image space).

Table 8 reports the retrieval performance across various datasets. SeMoBridge consistently outperforms the standard CLIP intra-modal baseline. Significant improvements are observed in finegrained datasets such as OxfordPets, Flowers102, and DTD. This confirms that our method preserves and effectively utilizes fine-grained visual details during the modality translation.

Table 8: Image-to-Image Retrieval performance.

| Method | OxfordPets | Flowers102 | FGVCAircraft | DTD | EuroSAT | StanfordCars | SUN397 | Caltech101 | UCF101 | Avg. |
|---|---|---|---|---|---|---|---|---|---|---|
| CLIP intra-modal | 36.27 | 70.81 | 19.04 | 30.69 | **51.22** | 31.00 | 35.88 | 80.83 | 49.83 | 45.06 |
| SeMoBridge-T (Fast Update) | **36.96** | **74.30** | **19.54** | **34.48** | 51.21 | **34.35** | **37.70** | **82.78** | **52.54** | **47.10** |

## A.8 TEXT-TEXT RETRIEVAL EXPERIMENTS

To test the bidirectional capability of our approach, we evaluate a Text-to-Image variant of SeMoBridge on text-text retrieval tasks. In this scenario, the goal is to retrieve relevant text documents given a text query.

Standard approaches compare text embeddings directly (Intra-modal Text-Text). Instead, we train a reverse SeMoBridge to map the text query into the image modality. This enables an inter-modal comparison between the bridged query (now in image space) and the gallery texts (in text space).

Table 9 presents the results across standard NLP retrieval benchmarks. SeMoBridge demonstrates superior performance compared to both the CLIP intra-modal baseline and the optimization-based method Cross The Gap (OVI) (Mistretta et al., 2025). This indicates that the modality gap affects both modalities symmetrically and that SeMoBridge effectively resolves this misalignment in both directions.

Table 9: Text-to-Text Retrieval Performance.

| Method | IMDB | 20News | Climate | DBPedia | FEVER | NFCorpus | NQ | SciDocs | SciFact | Avg. |
|---|---|---|---|---|---|---|---|---|---|---|
| CLIP intra-modal | 52.22 | 19.24 | 11.19 | 30.32 | 58.44 | 8.90 | 23.31 | 13.54 | 26.25 | 27.05 |
| Cross The Gap (OVI) | 52.30 | 33.10 | 15.30 | 39.10 | 70.50 | 12.20 | 33.60 | 16.80 | 33.20 | 34.01 |
| SeMoBridge Text2Img | 52.81 | 41.99 | **23.38** | **43.82** | 75.78 | 13.19 | 37.95 | 18.09 | 37.99 | 38.33 |
| SeMoBridge-T (Fast Update) Text2Img | **57.42** | **47.99** | 21.11 | 43.58 | **76.56** | **14.02** | **41.05** | **19.33** | **40.98** | **40.23** |

## A.9 Transferability Experiments

To assess whether SeMoBridge learns a domain-general semantic alignment rather than dataset-specific statistics, we performed a cross-dataset transfer experiment. We fine-tune the bridge parameters on ImageNet's few-shot splits (1–16 shots) and evaluate the resulting model directly on the other 10 downstream datasets without any further training.

Crucially, we disable the CSB term during this process. This ensures the bridge does not learn ImageNet-specific classification boundaries. Instead, it is forced to learn a global geometric projection that aligns the image modality with the text modality.

As shown in Table 10, the ImageNet-trained variant consistently improves accuracy across target datasets compared to the training-free baseline, despite never having seen the target domains.

This indicates that the intra-modal misalignment in CLIP is relatively consistent across different visual domains. SeMoBridge effectively captures this structural relationship, functioning as a robust plug-and-play module.

Table 10: Dataset-transfer evaluation with ViT-B/16. SeMoBridge-T$^\dagger$ denotes the variant where the bridge is fine-tuned on ImageNet few-shot and then transferred to other datasets.

| Method | Shots | ImageNet | OxfordPets | Flowers102 | FGVCAircraft | DTD | EuroSAT | StanfordCars | Food101 | SUN397 | Caltech101 | UCF101 | Avg. |
|---|---|---|---|---|---|---|---|---|---|---|---|---|---|
| SeMoBridge | 1 | 70.28±0.05 | 91.79±0.19 | 84.06±0.64 | 30.40±0.92 | 56.89±0.69 | 68.18±4.30 | 68.15±0.66 | 86.42±0.03 | 70.24±0.34 | 94.36±0.28 | 73.95±1.24 | 72.25 |
| SeMoBridge-T$^\dagger$ | 1 | 70.67±0.03 | 92.12±0.10 | 85.09±0.70 | 29.60±0.47 | 55.52±0.64 | 66.78±0.92 | 68.15±0.62 | 86.47±0.06 | 70.56±0.37 | 94.55±0.20 | 72.39±0.37 | 71.99 |
| SeMoBridge-T | 1 | 70.88±0.09 | 92.22±0.19 | 89.84±0.85 | 32.94±0.43 | 59.79±0.64 | 69.69±5.48 | 70.27±0.44 | 86.62±0.04 | 71.17±0.21 | 94.85±0.20 | 75.87±0.05 | 74.01 |
| SeMoBridge | 2 | 70.66±0.11 | 91.61±0.49 | 89.21±0.50 | 32.34±0.26 | 59.47±1.10 | 70.67±0.89 | 69.88±0.15 | 86.55±0.01 | 71.19±0.05 | 94.85±0.20 | 76.60±1.40 | 73.91 |
| SeMoBridge-T$^\dagger$ | 2 | 71.28±0.12 | 92.22±0.18 | 90.58±0.75 | 32.42±0.22 | 59.14±0.62 | 68.99±2.00 | 71.78±0.97 | 86.57±0.03 | 71.59±0.25 | 94.86±0.27 | 76.91±0.67 | 74.21 |
| SeMoBridge-T | 2 | 71.40±0.04 | 92.24±0.22 | 92.03±0.65 | 35.28±0.71 | 61.90±1.09 | 78.65±2.96 | 73.46±0.75 | 86.85±0.09 | 72.89±0.20 | 94.99±0.42 | 78.55±0.77 | 76.20 |
| SeMoBridge | 4 | 71.02±0.05 | 91.87±0.10 | 91.46±0.67 | 34.85±0.57 | 62.49±0.67 | 76.46±2.90 | 71.89±0.61 | 86.70±0.10 | 72.83±0.18 | 95.19±0.24 | 78.73±0.35 | 75.77 |
| SeMoBridge-T$^\dagger$ | 4 | 72.04±0.04 | 92.60±0.44 | 93.91±0.23 | 35.67±0.59 | 64.60±0.78 | 77.94±1.53 | 75.39±0.43 | 86.67±0.08 | 72.87±0.16 | 95.20±0.28 | 80.78±0.30 | 77.06 |
| SeMoBridge-T | 4 | 72.17±0.07 | 93.04±0.28 | 94.60±0.25 | 38.35±0.48 | 65.74±0.97 | 81.66±1.00 | 76.61±0.32 | 87.00±0.07 | 74.47±0.23 | 95.50±0.17 | 81.12±0.31 | 78.21 |
| SeMoBridge | 8 | 71.53±0.05 | 92.02±0.09 | 94.14±0.64 | 36.81±0.55 | 65.59±0.31 | 76.48±0.81 | 73.42±0.14 | 86.85±0.11 | 74.03±0.15 | 95.60±0.42 | 80.58±0.38 | 77.00 |
| SeMoBridge-T$^\dagger$ | 8 | 73.11±0.10 | 93.21±0.43 | 96.02±0.50 | 39.24±0.52 | 66.65±1.00 | 76.88±1.98 | 77.03±0.08 | 87.06±0.04 | 74.58±0.35 | 95.86±0.12 | 82.48±0.80 | 78.37 |
| SeMoBridge-T | 8 | 73.07±0.08 | 93.06±0.33 | 96.29±0.24 | 42.60±0.59 | 69.40±0.18 | 84.29±1.16 | 80.03±0.59 | 87.32±0.18 | 76.15±0.18 | 95.83±0.29 | 83.08±0.70 | 80.10 |
| SeMoBridge | 16 | 71.86±0.09 | 92.04±0.19 | 95.22±0.14 | 39.18±0.47 | 66.27±0.94 | 78.60±0.45 | 76.33±0.13 | 86.96±0.07 | 74.66±0.17 | 95.85±0.08 | 81.99±0.35 | 78.09 |
| SeMoBridge-T$^\dagger$ | 16 | 73.96±0.22 | 92.75±0.49 | 95.74±0.37 | 40.33±0.38 | 68.74±1.47 | 79.56±1.34 | 79.61±0.46 | 87.05±0.08 | 75.28±0.34 | 96.02±0.20 | 83.15±0.29 | 79.29 |
| SeMoBridge-T | 16 | 73.98±0.05 | 93.42±0.44 | 97.27±0.45 | 47.84±0.63 | 73.01±0.15 | 89.25±0.25 | 83.75±0.33 | 87.52±0.08 | 76.96±0.12 | 96.26±0.09 | 84.93±0.35 | 82.20 |

## A.10 SeMoBridge on SLIP

To verify that our proposed method is not tied to standard CLIP models, we evaluate SeMoBridge on the SLIP (Self-supervision meets Language-Image Pre-training) (Mu et al., 2022) framework. SLIP adds to CLIP's objective by adding a self-supervised contrastive loss (SimCLR) to the image branch.

Table 11 presents the performance of SeMoBridge and SeMoBridge-T on SLIP across all 11 datasets. Despite the differences in training, SeMoBridge-T consistently outperforms the training-free baseline across all shot settings.

Table 11: SeMoBridge on SLIP ViT-B/16.

| Method | Shots | Pets | Flowers | Aircraft | DTD | EuroSAT | Cars | Food101 | SUN397 | Caltech | UCF101 | ImageNet | Avg. |
|---|---|---|---|---|---|---|---|---|---|---|---|---|---|
| SeMoBridge | 1 | 45.13±0.86 | 80.97±1.13 | 14.61±0.32 | 48.35±1.52 | 61.64±3.73 | 12.78±0.16 | 66.78±0.09 | 61.76±0.11 | 86.61±0.58 | 54.89±0.86 | 50.44±0.08 | 53.09 |
| SeMoBridge-T | 1 | 47.64±0.36 | 83.78±0.53 | 16.43±0.49 | 49.41±1.10 | 66.10±6.45 | 15.25±0.14 | 67.81±0.28 | 63.13±0.44 | 87.75±0.46 | 58.45±0.84 | 51.67±0.05 | 55.22 |
| SeMoBridge | 2 | 49.43±0.05 | 87.86±0.67 | 17.26±0.97 | 53.66±1.06 | 63.86±3.73 | 15.06±0.32 | 67.45±0.13 | 63.77±0.25 | 88.26±0.74 | 61.93±0.64 | 51.48±0.26 | 56.37 |
| SeMoBridge-T | 2 | 53.04±1.57 | 89.48±0.68 | 19.99±0.73 | 53.70±1.35 | 73.07±1.89 | 19.47±0.08 | 68.51±0.10 | 65.87±0.29 | 89.21±0.32 | 64.15±0.40 | 53.30±0.18 | 59.07 |
| SeMoBridge | 4 | 53.26±0.56 | 92.89±0.63 | 18.01±0.48 | 57.72±0.88 | 76.16±1.00 | 18.09±0.16 | 68.16±0.10 | 66.90±0.02 | 89.06±0.20 | 67.01±0.56 | 53.14±0.01 | 60.04 |
| SeMoBridge-T | 4 | 56.72±1.09 | 93.11±0.15 | 22.05±1.25 | 59.56±0.96 | 79.49±1.63 | 24.44±0.37 | 69.80±0.03 | 69.13±0.24 | 90.33±0.14 | 69.59±0.80 | 55.23±0.03 | 62.68 |
| SeMoBridge | 8 | 57.44±0.52 | 95.16±0.19 | 21.81±1.10 | 62.69±0.18 | 79.50±1.37 | 22.15±0.33 | 69.73±0.17 | 69.50±0.07 | 90.40±0.14 | 71.42±0.78 | 54.89±0.28 | 63.15 |
| SeMoBridge-T | 8 | 62.02±0.56 | 95.46±0.05 | 26.96±0.30 | 63.00±0.12 | 82.94±3.00 | 30.98±0.19 | 71.88±0.26 | 71.55±0.46 | 91.46±0.31 | 73.61±0.96 | 57.63±0.20 | 66.14 |
| SeMoBridge | 16 | 61.20±0.94 | 95.81±0.42 | 25.15±0.84 | 65.62±0.85 | 81.53±0.72 | 25.65±0.19 | 70.65±0.12 | 71.33±0.06 | 91.20±0.26 | 74.69±0.74 | 56.94±0.09 | 65.43 |
| SeMoBridge-T | 16 | 65.02±0.40 | 96.57±0.27 | 34.30±0.20 | 67.81±0.77 | 88.86±0.31 | 37.74±0.36 | 73.41±0.20 | 73.64±0.17 | 92.71±0.17 | 78.20±0.48 | 60.37±0.13 | 69.88 |

## B  RANK CONSTRAINTS ON $\mathbf{W}_{\text{txt}}^{+}$

To investigate the geometric complexity of the modality gap, we analyze the performance of Se-MoBridge when the rank of the projection matrix $\mathbf{W}_{\text{txt}}^{+}$ is constrained. If the relationship between the image and text modalities is highly complex, a high-rank transformation would be necessary to capture it. Conversely, if the gap is a simple geometric shift, a low-rank transformation should suffice.

We apply Singular Value Decomposition (SVD) to the learned bridge matrix and truncate the singular values to retain only the top $k$ components (e.g., rank 256 and 128 for a 512-dimensional space).

As shown in Table 12, constraining the rank of the bridge does not lead to a drop in performance. Even when the rank is reduced to 128 (25% of full rank), the average accuracy remains comparable to the full-rank baseline.

Table 12: Effect of Rank Constraints on SeMoBridge's $\mathbf{W}_{\text{txt}}^{+}$.

| | | Number of Shots | | | | |
|---|---|---|---|---|---|---|
| Constraint | 1 | 2 | 4 | 8 | 16 | Avg. |
| Full Rank (512) | 72.25 | 73.91 | 75.77 | 77.00 | 78.09 | 75.40 |
| Rank ½ (256) | 72.33 | 73.90 | 75.79 | 76.98 | 78.07 | 75.41 |
| Rank ¼ (128) | 72.23 | 74.09 | 75.76 | 76.95 | 78.30 | 75.47 |

## C  LLM USAGE FOR WRITING OF THIS PAPER

LLMs were used as a writing aid throughout the preparation of this manuscript. We employed LLMs to assist with sentence formulation, improve clarity, and for general grammatical polishing to refine the overall readability of the text.

