# OpenReview forum: "SeMoBridge: Semantic Modality Bridge for Efficient Few-Shot Adaptation of CLIP"
_ICLR.cc/2026/Conference — Submitted to ICLR 2026_

### Official Review · Reviewer_WGAS · 2025-10-28

**Soundness:** 3
**Presentation:** 3
**Contribution:** 3
**Rating:** 6
**Confidence:** 3

**Summary:**

The paper pinpoints an overlooked limitation of CLIP in few-shot classification—namely intra-modal misalignment caused by the modality gap—and proposes SeMoBridge, a lightweight projection that maps image embeddings into the text modality. A closed-form, training-free version and an optionally fine-tuned variant (SeMoBridge-T) are introduced; the latter is trained with a multi-modal loss while keeping CLIP frozen. Experiments show consistent state-of-the-art accuracy in 1/2/4-shot regimes on several benchmarks with 5-10× less computation than prior methods such as Cross-the-Gap.

**Strengths:**

(1) Clear motivation and visual evidence of intra-modal misalignment; (2) Simple global projection applicable to all inputs, avoiding per-sample optimisation; (3) Efficient fine-tuning that never back-propagates through CLIP encoders; (4) Strong empirical gains and detailed ablations; (5) Low engineering overhead, making the method practical.

**Weaknesses:**

(1) Theoretical justification is limited; why a single linear map suffices is not deeply analysed; (2) Evaluation is restricted to image classification with ViT-B/16; effect on larger backbones, other modalities, and tasks (retrieval, detection) is unknown; (3) Only image-to-text projection studied—no comparison with text-to-image or shared latent bridges; (4) Dependence on prompt engineering is unclear; (5) Robustness across seeds and potential degradation of zero-shot performance are not reported.

**Questions:**

(Q1) What exact closed-form solution is used—Procrustes on class centroids or something else? (Q2) Does constraining the projection’s rank change results? (Q3) What happens if both query and support images are projected before comparison? (Q4) Can a bridge trained on one dataset transfer to another without re-tuning? (Q5) How much does SeMoBridge affect original CLIP zero-shot accuracy?

---

> ### Author Response · Authors · 2025-11-21
>
> We thank the reviewer for their time and their helpful feedback on our submission. We believe that it provides valuable perspectives that will enhance the quality and clarity of our work.
>
> **Below, we provide detailed responses addressing each of the weaknesses [W#] and questions [Q#] raised.**
>
> ---
>
> ## [W1] Theoretical Justification for Linear Projection
> We appreciate you raising this point. **We argue that a linear projection is the most theoretically grounded baseline for this specific architecture.**
>
> The CLIP text encoder produces final text embeddings via a linear projection $\mathbf{W}\_{\mathrm{txt}}$ applied to the EOS (end-of-sentence) token representation. Because this projection is the transformation that CLIP actually learns during pre-training, the resulting text embedding space defines the semantic directions that matching images and texts are encouraged to align with.
>
> CLIP’s contrastive learning objective optimizes normalized image and text embeddings to point in the same direction on the hypersphere:
> $$
> \frac{\mathbf{f}\_{\mathrm{img}}}{\lVert \mathbf{f}\_{\mathrm{img}} \rVert}
> \approx
> \frac{\mathbf{f}\_{\mathrm{txt}}}{\lVert \mathbf{f}\_{\mathrm{txt}} \rVert}.
> $$
> Since $\mathbf{f}\_{\mathrm{txt}} = \mathbf{W}\_{\mathrm{txt}}\,\mathbf{f}\_{\mathrm{eos}}$, the text space already encodes the “correct” semantic direction determined by CLIP training. Therefore, when constructing a *bridged* text feature $\hat{\mathbf{f}}\_{\mathrm{txt}}$, it is natural to require that its direction matches that of $\mathbf{f}\_{\mathrm{img}}$.
>
> To achieve this, SeMoBridge uses the Moore–Penrose pseudo-inverse of the text projection. Because the forward mapping is linear, its structural inverse is also linear. As noted in Section 3.2 (Eq. 3), the composition
> $$
> \mathbf{W}\_{\mathrm{txt}} \, \mathbf{W}\_{\mathrm{txt}}^{+} \approx \mathrm{I}
> $$
>
> acts approximately as an identity transformation in the text embedding space.
>
> As a result, if we estimate a pseudo-EOS feature via the bridge,
>
> $$
> \hat{\mathbf{f}}\_{\mathrm{eos}} \approx \mathbf{W}\_{\mathrm{txt}}^{+}\,\mathbf{f}\_{\mathrm{img}},
> $$
>
> the resulting bridged text feature becomes:
>
> $$
> \hat{\mathbf{f}}\_{\mathrm{txt}} =
> \mathbf{W}\_{\mathrm{txt}}\,\hat{\mathbf{f}}\_{\mathrm{eos}}
> \approx
> \mathbf{W}\_{\mathrm{txt}}\mathbf{W}\_{\mathrm{txt}}^{+}\mathbf{f}\_{\mathrm{img}}
> \approx
> \mathbf{f}\_{\mathrm{img}}.
> $$
>
> In other words, all hat-annotated features refer to *bridge-estimated* text-side representations, and their alignment with $\mathbf{f}_{\mathrm{img}}$ is a direct consequence of CLIP’s original cosine-similarity pre-training. This reinforces why a linear bridge is not only theoretically justified but also an architecturally faithful inverse of CLIP’s design.

---

> ### Author Response · Authors · 2025-11-21
>
> ## [W2] Generalization across Downstream Tasks and Datasets (1/2)
> Thank you for suggesting more experiments.
>
> **Leading methods in few-shot CLIP adaptation (e.g., CoOp, Tip-Adapter, Cross-the-Gap) primarily report results using ResNet-50 and ViT-B/16** as standard benchmarks. Therefore, **we followed this to ensure our results are directly comparable**. However, we acknowledge the importance of validating scalability on larger models. If the reviewer considers this critical for the final assessment, **we are fully prepared to conduct additional experiments using larger backbones (e.g., ViT-L/14)** and report the results during the discussion period.
>
> To validate the versatility of SeMoBridge, **we conducted additional experiments on Retrieval (both Image-Image and Text-Text)**, following the standard evaluation setting used in related works like Cross the Gap (Mistretta et al., 2025).
>
> **Image-Image Retrieval (Bridging Image to Text modality):** We applied SeMoBridge to project query images into the text modality for inter-modal retrieval (picking the most similar $k$ images from a gallery).
>
> |**Model**|**Method**|**OxfordPets**|**Flowers102**|**FGVCAircraft**|**DTD**|**EuroSAT**|**StanfordCars**|**Food101**|**SUN397**|**Caltech101**|**UCF101**|**ImageNet**|**AVERAGE**|
> |---|---|---|---|---|---|---|---|---|---|---|---|---|---|
> |ViT-B/16|CLIP intra-modal|36.27|70.81|19.04|30.69|**51.22**|31.00|—|35.88|80.83|49.83|—|45.06|
> |ViT-B/16|SeMoBridge-T (Fast Update)|**36.96**|**74.30**|**19.54**|**34.48**|51.21|**34.35**|—|**37.70**|**82.78**|**52.54**|—|**47.10**|
>
> (Food101 and ImageNet results are missing due to VRAM issues.)
>
>
> **Text-Text Retrieval (Bridging Text to Image modality):** We also experimented with a **Text-to-Image version of SeMoBridge**, to map text queries into the image modality to retrieve similar texts.
> |**Model**|**Method**|**IMDB_Reviews**|**20News.**|**Climate**|**DBPedia**|**FEVER**|**NFCorpus**|**NQ**|**SciDocs**|**SciFact**|**AVERAGE**|
> |---|---|---|---|---|---|---|---|---|---|---|---|
> |ViT-B/32|CLIP intra-modal|52.22|19.24|11.19|30.32|58.44|8.90|23.31|13.54|26.25|27.05|
> |ViT-B/32|Cross The Gap (OVI)|52.30|33.10|15.30|39.10|70.50|12.20|33.60|16.80|33.20|34.01|
> |ViT-B/32|SeMoBridge Text2Img|52.81|41.99|**23.38**|**43.82**|75.78|13.19|37.95|18.09|37.99|38.33|
> |ViT-B/32|SeMoBridge-T (Fast Update) Text2Img|**57.42**|**47.99**|21.11|43.58|**76.56**|**14.02**|**41.05**|**19.33**|**40.98**|**40.23**|
>
> As shown in the results, **SeMoBridge consistently improves performance in both Image-to-Text and Text-to-Image retrieval tasks**.

---

> ### Author Response · Authors · 2025-11-21
>
> ## [W2] Generalization across Downstream Tasks and Datasets (2/2)
>
> Additionally, **we explored SeMoBridge’s generalizability further by fine-tuning the bridge on ImageNet’s few-shot set (1-16 shots), and then evaluating it on other datasets (dataset transfer).**
>
> |Model|Method|Shots|ImageNet|OxfordPets|Flowers102|FGVCAircraft|DTD|EuroSAT|StanfordCars|Food101|SUN397|Caltech101|UCF101|AVERAGE|
> |---|---|---|---|---|---|---|---|---|---|---|---|---|---|---|
> |ViT-B/16|CLIP zero-shot|0|68.73±0.00|89.10±0.00|70.73±0.00|24.69±0.00|44.09±0.00|48.31±0.00|65.61±0.00|85.87±0.00|62.59±0.00|93.35±0.00|67.62±0.00|65.52|
> |ViT-B/16|SeMoBridge|1|70.28±0.05|91.79±0.19|84.06±0.64|30.40±0.92|56.89±0.69|68.18±4.30|68.15±0.66|86.42±0.03|70.24±0.34|94.36±0.28|73.95±1.24|72.25|
> |ViT-B/16|SeMoBridge-T (ImageNet-Trained)|1|70.67±0.03|92.12±0.10|85.09±0.70|29.60±0.47|55.52±0.64|66.78±0.92|68.15±0.62|86.47±0.06|70.56±0.37|94.55±0.20|72.39±0.37|71.99|
> |ViT-B/16|SeMoBridge-T|1|70.88±0.09|92.22±0.19|89.84±0.85|32.94±0.43|59.79±0.64|69.69±5.48|70.27±0.44|86.62±0.04|71.17±0.21|94.85±0.20|75.87±0.56|74.01|
> |ViT-B/16|SeMoBridge|2|70.66±0.11|91.61±0.49|89.21±0.50|32.34±0.26|59.47±1.10|70.67±0.89|69.88±0.15|86.55±0.01|71.19±0.05|94.85±0.20|76.60±1.40|73.91|
> |ViT-B/16|SeMoBridge-T (ImageNet-Trained)|2|71.28±0.12|92.22±0.18|90.58±0.75|32.42±0.22|59.14±0.62|68.99±2.00|71.78±0.97|86.57±0.03|71.59±0.25|94.86±0.27|76.91±0.67|74.21|
> |ViT-B/16|SeMoBridge-T|2|71.40±0.04|92.24±0.22|92.03±0.65|35.28±0.71|61.90±1.09|78.65±2.96|73.46±0.75|86.85±0.09|72.89±0.20|94.99±0.42|78.55±0.77|76.20|
> |ViT-B/16|SeMoBridge|4|71.02±0.05|91.87±0.10|91.46±0.67|34.85±0.57|62.49±0.67|76.46±2.90|71.89±0.61|86.70±0.10|72.83±0.18|95.19±0.24|78.73±0.35|75.77|
> |ViT-B/16|SeMoBridge-T (ImageNet-Trained)|4|72.04±0.04|92.60±0.44|93.91±0.23|35.67±0.59|64.60±0.78|77.94±1.53|75.39±0.43|86.67±0.08|72.87±0.16|95.20±0.28|80.78±0.30|77.06|
> |ViT-B/16|SeMoBridge-T|4|72.17±0.07|93.04±0.28|94.60±0.25|38.35±0.48|65.74±0.97|81.66±1.00|76.61±0.32|87.00±0.07|74.47±0.23|95.50±0.17|81.12±0.31|78.21|
> |ViT-B/16|SeMoBridge|8|71.53±0.05|92.02±0.09|94.14±0.64|36.81±0.55|65.59±0.31|76.48±0.81|73.42±0.14|86.85±0.11|74.03±0.15|95.60±0.42|80.58±0.38|77.00|
> |ViT-B/16|SeMoBridge-T (ImageNet-Trained)|8|73.11±0.10|93.21±0.43|96.02±0.50|39.24±0.52|66.65±1.00|76.88±1.98|77.03±0.88|87.06±0.04|74.58±0.35|95.86±0.12|82.48±0.80|78.37|
> |ViT-B/16|SeMoBridge-T|8|73.07±0.08|93.06±0.33|96.29±0.24|42.60±0.59|69.40±0.18|84.29±1.16|80.03±0.59|87.32±0.18|76.15±0.18|95.83±0.29|83.08±0.70|80.10|
> |ViT-B/16|SeMoBridge|16|71.86±0.09|92.04±0.19|95.22±0.14|39.18±0.47|66.27±0.94|78.60±0.45|76.33±0.13|86.96±0.07|74.66±0.17|95.85±0.08|81.99±0.35|78.09|
> |ViT-B/16|SeMoBridge-T (ImageNet-Trained)|16|73.96±0.22|92.75±0.49|95.74±0.37|40.33±0.38|68.74±1.47|79.56±1.34|79.61±0.46|87.05±0.08|75.28±0.34|96.02±0.20|83.15±0.29|79.29|
> |ViT-B/16|SeMoBridge-T|16|73.98±0.05|93.42±0.44|97.27±0.45|47.84±0.63|73.01±0.15|89.25±0.25|83.75±0.33|87.52±0.08|76.96±0.12|96.26±0.09|84.93±0.35|82.20|
>
> As shown in the Table, **fine-tuning on ImageNet also improves performance for different domains**.
>
> This confirms that **our method effectively learns general semantic alignment, reinforcing the claim that SeMoBridge successfully resolves intra-modal misalignment across different domains.** It makes it a highly practical "plug-and-play" module that can be deployed across various tasks without dataset-specific re-training.

---

> ### Author Response · Authors · 2025-11-21
>
> ## [W3] Text-to-Image Projection
> Thank you for suggesting this. We clarify that while our primary focus was on image classification (Image-to-Text), **we have investigated the Text-to-Image direction** to validate the universality of our approach.
>
> As detailed in our response to **[W2]**, **we have successfully applied SeMoBridge in the Text-to-Image direction for retrieval tasks**. The results confirm that **our projection method works effectively in the reverse direction as well**, outperforming Cross the Gap (Mistretta et al., 2025).

---

> ### Author Response · Authors · 2025-11-21
>
> ## [W4] Dependency on Text Descriptions
> Thank you for asking about this. We clarify that **our method does not depend on custom prompts to function effectively**, but rather benefits from them in cases where image supervision is not enough.
>
> |**Prompt Template**|**zero-shot CLIP**|**1 shot**|**2 shots**|**4 shots**|**8 shots**|**16 shots**|
> |---|---|---|---|---|---|---|
> |**"a photo of a {}"**|63.79|71.73|74.96|77.42|79.56|81.90|
> |**CLIP prompts**|62.93|72.10|75.11|77.42|79.71|81.83|
> |**CLIP ensemble**|65.87|72.31|75.23|77.63|79.73|82.01|
> |**CuPL**|68.66|73.88|76.08|78.20|80.07|82.19|
> |**CLIP ensemble + CuPL**|68.91|74.01|76.20|78.21|80.10|82.20|
>
> *Figure 6 (right) is here:* https://imgur.com/2B5Xxnj
>
> As shown in the table (**Figure 6 (right) in the paper**), we have provided an ablation study addressing this concern. SeMoBridge-T achieves strong performance (significantly outperforming baselines) even when using simple, generic prompts like “a photo of a {}”. This proves that the quality of textual descriptions is not a bottleneck.
>
> The performance gain with LLM-generated prompts (e.g., CuPL (Pratt et al., 2023)) demonstrates our method's extensibility, not a dependency.
> By leveraging the richer textual information provided by these prompts, SeMoBridge performs a more precise transformation of image embeddings into the text modality, especially when there are very few images to work with (1-2 shots). **When more images are available (4-16 shots), the difference between generic and LLM-generated texts becomes very small.** Our method is capable of fully exploiting available semantic information from both image and text to maximize performance, while remaining effective with generic prompts.
>
> **Using CuPL prompts is not specific to us**; it follows the standard setting of recent state-of-the-art methods (e.g., APE (Zhu et al., 2023)) to ensure fair comparison.

---

> ### Author Response · Authors · 2025-11-21
>
> ## [W5] Zero-Shot Performance and Robustness
> Thank you for bringing this up. **There is no degradation of the original zero-shot performance.** Unlike full fine-tuning methods that modify the model weights (risking catastrophic forgetting), SeMoBridge keeps the original CLIP encoders and projections **frozen.**
>
> To ensure reproducibility and stability, **all our reported results are averaged over 3 different random seeds** (as in CoOp (Zhou et al., 2022)).
>
> In the following table, all results are are formatted as XX.XX (Accuracy) ± X.XX (standard deviation)
>
> |Model|Method|Shots|ImageNet|OxfordPets|Flowers102|FGVCAircraft|DTD|EuroSAT|StanfordCars|Food101|SUN397|Caltech101|UCF101|AVERAGE|
> |---|---|---|---|---|---|---|---|---|---|---|---|---|---|---|
> |ViT-B/16|CLIP zero-shot|0|68.73±0.00|89.10±0.00|70.73±0.00|24.69±0.00|44.09±0.00|48.31±0.00|65.61±0.00|85.87±0.00|62.59±0.00|93.35±0.00|67.62±0.00|65.52|
> |ViT-B/16|SeMoBridge|1|70.28±0.05|91.79±0.19|84.06±0.64|30.40±0.92|56.89±0.69|68.18±4.30|68.15±0.66|86.42±0.03|70.24±0.34|94.36±0.28|73.95±1.24|72.25|
> |ViT-B/16|SeMoBridge-T|1|70.88±0.09|92.22±0.19|89.84±0.85|32.94±0.43|59.79±0.64|69.69±5.48|70.27±0.44|86.62±0.04|71.17±0.21|94.85±0.20|75.87±0.56|74.01|
> |ViT-B/16|SeMoBridge|2|70.66±0.11|91.61±0.49|89.21±0.50|32.34±0.26|59.47±1.10|70.67±0.89|69.88±0.15|86.55±0.01|71.19±0.05|94.85±0.20|76.60±1.40|73.91|
> |ViT-B/16|SeMoBridge-T|2|71.40±0.04|92.24±0.22|92.03±0.65|35.28±0.71|61.90±1.09|78.65±2.96|73.46±0.75|86.85±0.09|72.89±0.20|94.99±0.42|78.55±0.77|76.20|
> |ViT-B/16|SeMoBridge|4|71.02±0.05|91.87±0.10|91.46±0.67|34.85±0.57|62.49±0.67|76.46±2.90|71.89±0.61|86.70±0.10|72.83±0.18|95.19±0.24|78.73±0.35|75.77|
> |ViT-B/16|SeMoBridge-T|4|72.17±0.07|93.04±0.28|94.60±0.25|38.35±0.48|65.74±0.97|81.66±1.00|76.61±0.32|87.00±0.07|74.47±0.23|95.50±0.17|81.12±0.31|78.21|
> |ViT-B/16|SeMoBridge|8|71.53±0.05|92.02±0.09|94.14±0.64|36.81±0.55|65.59±0.31|76.48±0.81|73.42±0.14|86.85±0.11|74.03±0.15|95.60±0.42|80.58±0.38|77.00|
> |ViT-B/16|SeMoBridge-T|8|73.07±0.08|93.06±0.33|96.29±0.24|42.60±0.59|69.40±0.18|84.29±1.16|80.03±0.59|87.32±0.18|76.15±0.18|95.83±0.29|83.08±0.70|80.10|
> |ViT-B/16|SeMoBridge|16|71.86±0.09|92.04±0.19|95.22±0.14|39.18±0.47|66.27±0.94|78.60±0.45|76.33±0.13|86.96±0.07|74.66±0.17|95.85±0.08|81.99±0.35|78.09|
> |ViT-B/16|SeMoBridge-T|16|73.98±0.05|93.42±0.44|97.27±0.45|47.84±0.63|73.01±0.15|89.25±0.25|83.75±0.33|87.52±0.08|76.96±0.12|96.26±0.09|84.93±0.35|82.20|
>
> As shown here, **the deviation across seeds is very low, confirming that SeMoBridge provides consistent improvements regardless of random initialization or support set sampling.**

---

> ### Author Response · Authors · 2025-11-21
>
> ## [Q1] Closed-Form Solution
> Thank you for the question. We use the **Moore-Penrose pseudoinverse** to solve for an unconstrained linear projection, as detailed in Section 3.2.
>
> If we did not understand the question correctly, please let us know.

---

> ### Author Response · Authors · 2025-11-21
>
> ## [Q2] Rank Constraints on $\mathbf{W}_{\mathrm{txt}}^+$
> This is a very interesting question.
> To investigate this, we performed experiments on both training-free SeMoBridge and trained SeMoBridge-T on ViT-B/16 with $\mathbf{W}_{\mathrm{txt}}^+ \in \mathbb R^{ 512 \times 512}$.
>
> We performed the experiment as suggested, constraining the rank to ½ (256) and ¼ (128) via SVD truncation.
>
> |**Rank constraint**|**1 shot**|**2 shots**|**4 shots**|**8 shots**|**16 shots**|**AVERAGE**|
> |---|---:|---:|---:|---:|---:|---:|
> |**Full rank (512)**|72.25|73.91|75.77|77.00|78.09|75.40|
> |**1/2 (256)**|72.33|73.90|75.79|76.98|78.07|75.41|
> |**1/4 (128)**|72.23|74.09|75.76|76.95|78.30|75.47|
>
> As shown in the table, even when we aggressively reduced the rank to ¼ (128), the performance did not drop.
>
> This result provides a profound insight. While intra-modal misalignment causes significant confusion in downstream tasks, the underlying modality gap causing it is geometrically simpler than expected.
>
> It proves that we can resolve the misalignment using a surprisingly lightweight (low-rank) transformation. This validates why our simple linear projection is not only sufficient but optimal; it captures the essential correction direction without carrying unnecessary complexity.

---

> ### Author Response · Authors · 2025-11-21
>
> ## [Q3] Comparing after Projecting both Query and Support images
> To answer this interesting question, **we conducted experiments to show intra-modal misalignment quantitatively**. We calculated the “overlap” (or confusion) between unpaired and paired image cosine similaritiy distributions.
>
> |**Comparison Method**|**Unpaired/Paired overlap**|
> |---|---:|
> |Intra-Modal Image-Image (No bridge)|22.99%|
> |Intra-Modal Image-Image (Both bridged)|23.20%|
> |SeMoBridge Image-Image (Only query bridged)|0.78%|
> |SeMoBridge-T Image-Image (Only query bridged)|**0.02%**|
>
> *A figure showing the cosine similarity distributions:* https://imgur.com/2CWC8EE
>
> This shows that **bridging both the query and support images (yellow) results in the same intra-modal misalignment problem** as standard image-image comparisons. **Only inter-modal comparisons are reliable**, due to CLIP’s alignment from inter-modal contrastive loss during pre-training.

---

> ### Author Response · Authors · 2025-11-21
>
> ## [Q4] Cross-Dataset Transferability
> This is a critical question regarding the generalization capability of our learned projection.
>
> **We explored SeMoBridge’s generalizability by fine-tuning the bridge on ImageNet’s few-shot set (1-16 shots), and then evaluating it on other datasets (dataset transfer).**
>
>
> |Model|Method|Shots|ImageNet|OxfordPets|Flowers102|FGVCAircraft|DTD|EuroSAT|StanfordCars|Food101|SUN397|Caltech101|UCF101|AVERAGE|
> |---|---|---|---|---|---|---|---|---|---|---|---|---|---|---|
> |ViT-B/16|CLIP zero-shot|0|68.73±0.00|89.10±0.00|70.73±0.00|24.69±0.00|44.09±0.00|48.31±0.00|65.61±0.00|85.87±0.00|62.59±0.00|93.35±0.00|67.62±0.00|65.52|
> |ViT-B/16|SeMoBridge|1|70.28±0.05|91.79±0.19|84.06±0.64|30.40±0.92|56.89±0.69|68.18±4.30|68.15±0.66|86.42±0.03|70.24±0.34|94.36±0.28|73.95±1.24|72.25|
> |ViT-B/16|SeMoBridge-T (ImageNet-Trained)|1|70.67±0.03|92.12±0.10|85.09±0.70|29.60±0.47|55.52±0.64|66.78±0.92|68.15±0.62|86.47±0.06|70.56±0.37|94.55±0.20|72.39±0.37|71.99|
> |ViT-B/16|SeMoBridge-T|1|70.88±0.09|92.22±0.19|89.84±0.85|32.94±0.43|59.79±0.64|69.69±5.48|70.27±0.44|86.62±0.04|71.17±0.21|94.85±0.20|75.87±0.56|74.01|
> |ViT-B/16|SeMoBridge|2|70.66±0.11|91.61±0.49|89.21±0.50|32.34±0.26|59.47±1.10|70.67±0.89|69.88±0.15|86.55±0.01|71.19±0.05|94.85±0.20|76.60±1.40|73.91|
> |ViT-B/16|SeMoBridge-T (ImageNet-Trained)|2|71.28±0.12|92.22±0.18|90.58±0.75|32.42±0.22|59.14±0.62|68.99±2.00|71.78±0.97|86.57±0.03|71.59±0.25|94.86±0.27|76.91±0.67|74.21|
> |ViT-B/16|SeMoBridge-T|2|71.40±0.04|92.24±0.22|92.03±0.65|35.28±0.71|61.90±1.09|78.65±2.96|73.46±0.75|86.85±0.09|72.89±0.20|94.99±0.42|78.55±0.77|76.20|
> |ViT-B/16|SeMoBridge|4|71.02±0.05|91.87±0.10|91.46±0.67|34.85±0.57|62.49±0.67|76.46±2.90|71.89±0.61|86.70±0.10|72.83±0.18|95.19±0.24|78.73±0.35|75.77|
> |ViT-B/16|SeMoBridge-T (ImageNet-Trained)|4|72.04±0.04|92.60±0.44|93.91±0.23|35.67±0.59|64.60±0.78|77.94±1.53|75.39±0.43|86.67±0.08|72.87±0.16|95.20±0.28|80.78±0.30|77.06|
> |ViT-B/16|SeMoBridge-T|4|72.17±0.07|93.04±0.28|94.60±0.25|38.35±0.48|65.74±0.97|81.66±1.00|76.61±0.32|87.00±0.07|74.47±0.23|95.50±0.17|81.12±0.31|78.21|
> |ViT-B/16|SeMoBridge|8|71.53±0.05|92.02±0.09|94.14±0.64|36.81±0.55|65.59±0.31|76.48±0.81|73.42±0.14|86.85±0.11|74.03±0.15|95.60±0.42|80.58±0.38|77.00|
> |ViT-B/16|SeMoBridge-T (ImageNet-Trained)|8|73.11±0.10|93.21±0.43|96.02±0.50|39.24±0.52|66.65±1.00|76.88±1.98|77.03±0.88|87.06±0.04|74.58±0.35|95.86±0.12|82.48±0.80|78.37|
> |ViT-B/16|SeMoBridge-T|8|73.07±0.08|93.06±0.33|96.29±0.24|42.60±0.59|69.40±0.18|84.29±1.16|80.03±0.59|87.32±0.18|76.15±0.18|95.83±0.29|83.08±0.70|80.10|
> |ViT-B/16|SeMoBridge|16|71.86±0.09|92.04±0.19|95.22±0.14|39.18±0.47|66.27±0.94|78.60±0.45|76.33±0.13|86.96±0.07|74.66±0.17|95.85±0.08|81.99±0.35|78.09|
> |ViT-B/16|SeMoBridge-T (ImageNet-Trained)|16|73.96±0.22|92.75±0.49|95.74±0.37|40.33±0.38|68.74±1.47|79.56±1.34|79.61±0.46|87.05±0.08|75.28±0.34|96.02±0.20|83.15±0.29|79.29|
> |ViT-B/16|SeMoBridge-T|16|73.98±0.05|93.42±0.44|97.27±0.45|47.84±0.63|73.01±0.15|89.25±0.25|83.75±0.33|87.52±0.08|76.96±0.12|96.26±0.09|84.93±0.35|82.20|
>
> As shown in the Table, **fine-tuning on ImageNet also improves performance for different domains**.
>
> This confirms that **our method effectively learns general semantic alignment, reinforcing the claim that SeMoBridge successfully resolves intra-modal misalignment across different domains.** It makes it a highly practical "plug-and-play" module that can be deployed across various tasks without dataset-specific re-training.

---

> ### Author Response · Authors · 2025-11-21
>
> ## [Q5] Impact on Original Zero-Shot Accuracy
> Thank you for your question. **SeMoBridge does not affect the original CLIP zero-shot accuracy**. Unlike fine-tuning methods that modify model weights and risk catastrophic forgetting, **we keep CLIP encoders and projections completely frozen**. We add two logits ($z_2$ and $z_3$) to the original zero-shot logits $z_1$ using the bridged few-shot images to improve classification accuracy.
>
> If we did not understand your question correctly, please let us know.

---

### Official Review · Reviewer_ay5F · 2025-10-30

**Soundness:** 3
**Presentation:** 3
**Contribution:** 3
**Rating:** 4
**Confidence:** 4

**Summary:**

The paper introduces SeMoBridge, a lightweight method designed to address the challenge of intra-modal misalignment in CLIP (Contrastive Language-Image Pretraining) when applied to few-shot classification tasks. CLIP, while highly effective in zero-shot settings, struggles with few-shot learning due to its inability to calibrate the image modality during training. The authors propose a Semantic Modality Bridge (SeMoBridge) to map images into the text modality, thus enabling more reliable comparisons within CLIP's shared embedding space. The method operates with minimal computational cost and offers significant improvements in accuracy, particularly in low-data settings (1-4 shots). A trained version of SeMoBridge (SeMoBridge-T) further enhances performance through multi-modal supervision. Extensive experiments across 11 datasets confirm that SeMoBridge outperforms state-of-the-art methods, especially in low-shot scenarios.

**Strengths:**

1. The proposed SeMoBridge effectively resolves the intra-modal misalignment that impacts CLIP's performance in few-shot learning tasks. The approach of mapping images into the text modality for alignment is both novel and practical.

2. SeMoBridge is training-free, reducing computational cost by eliminating the need for expensive per-sample optimization. Even in the trained version (SeMoBridge-T), the method requires minimal training time compared to existing alternatives, such as APE and Tip-Adapter.

3. The paper reports extensive experiments across various benchmarks, demonstrating SeMoBridge’s effectiveness in low-shot scenarios (1, 2, 4 shots). The experiments also show SeMoBridge’s robustness to distribution shifts, making it applicable to real-world settings.

**Weaknesses:**

1. While the method is demonstrated to work well on low-shot settings, there is a lack of evaluation on very large-scale datasets beyond the commonly used benchmarks like ImageNet, Caltech101, and others. Testing on more diverse datasets could provide a clearer picture of SeMoBridge's generalization capabilities.

2. The method heavily relies on the CLIP model for both image and text embeddings. While CLIP is a state-of-the-art model, this dependence limits the method's applicability to tasks where CLIP's pre-trained features may not be optimal or where fine-tuning CLIP is necessary.

3. SeMoBridge's mapping from image embeddings to the text modality, while effective for alignment, might potentially obscure fine-grained visual details in some cases, as it primarily leverages textual descriptions for image alignment.

4. The multi-modal supervision in SeMoBridge-T shows significant performance improvement, especially in low-data settings. However, the requirement for text supervision may not always be practical in real-world applications where text descriptions for every class might not be readily available.

**Questions:**

1. How does SeMoBridge handle scenarios where textual prompts or descriptions are unavailable or difficult to generate for specific classes? Would a zero-text version of SeMoBridge be possible, and how would it compare in performance?

2. How does SeMoBridge perform in settings with noisy or ambiguous class descriptions? Are there mechanisms in place to mitigate the potential issues arising from imperfect text prompts?

3. Can SeMoBridge be extended to multi-modal tasks beyond few-shot image classification, such as multi-modal retrieval or object detection? What are the implications of using SeMoBridge for tasks where visual details are more critical than text alignment?

---

> ### Author Response · Authors · 2025-11-21
>
> We thank the reviewer for their time and their helpful feedback on our submission. We believe that it provides valuable perspectives that will enhance the quality and clarity of our work.
>
> **Below, we provide detailed responses addressing each of the weaknesses [W#] and questions [Q#] raised.**
>
> ---
>
> ## [W1] Evaluation Scope
>
> Thank you for your suggestion. We respectfully clarify that **our experimental setting follows the standard evaluation established by prior state-of-the-art works in few-shot learning** (e.g., CoOp, Tip-Adapter, APE, SkipT, LDC).
>
> To ensure fair and direct comparisons, we adopted the widely accepted benchmark suite consisting of 11 diverse datasets. We argue that this sufficiently validates the generalization capabilities of SeMoBridge across varied domains.
>
> However, **to explore SeMoBridge’s generalizability further, we fine-tuned the bridge on ImageNet’s few-shot set (1–16 shots), and then evaluated it on other datasets** (dataset transfer).
>
> |Model|Method|Shots|ImageNet|OxfordPets|Flowers102|FGVCAircraft|DTD|EuroSAT|StanfordCars|Food101|SUN397|Caltech101|UCF101|AVERAGE|
> |---|---|---|---|---|---|---|---|---|---|---|---|---|---|---|
> |ViT-B/16|CLIP zero-shot|0|68.73±0.00|89.10±0.00|70.73±0.00|24.69±0.00|44.09±0.00|48.31±0.00|65.61±0.00|85.87±0.00|62.59±0.00|93.35±0.00|67.62±0.00|65.52|
> | ViT-B/16|SeMoBridge|1|70.28±0.05|91.79±0.19|84.06±0.64|30.40±0.92|56.89±0.69|68.18±4.30|68.15±0.66|86.42±0.03|70.24±0.34|94.36±0.28|73.95±1.24|72.25|
> | ViT-B/16|SeMoBridge-T (ImageNet-Trained)|1|70.67±0.03|92.12±0.10|85.09±0.70|29.60±0.47|55.52±0.64|66.78±0.92|68.15±0.62|86.47±0.06|70.56±0.37|94.55±0.20|72.39±0.37|71.99|
> | ViT-B/16|SeMoBridge-T|1|70.88±0.09|92.22±0.19|89.84±0.85|32.94±0.43|59.79±0.64|69.69±5.48|70.27±0.44|86.62±0.04|71.17±0.21|94.85±0.20|75.87±0.56|74.01|
> | ViT-B/16|SeMoBridge|2|70.66±0.11|91.61±0.49|89.21±0.50|32.34±0.26|59.47±1.10|70.67±0.89|69.88±0.15|86.55±0.01|71.19±0.05|94.85±0.20|76.60±1.40|73.91|
> | ViT-B/16|SeMoBridge-T (ImageNet-Trained)|2|71.28±0.12|92.22±0.18|90.58±0.75|32.42±0.22|59.14±0.62|68.99±2.00|71.78±0.97|86.57±0.03|71.59±0.25|94.86±0.27|76.91±0.67|74.21|
> | ViT-B/16|SeMoBridge-T|2|71.40±0.04|92.24±0.22|92.03±0.65|35.28±0.71|61.90±1.09|78.65±2.96|73.46±0.75|86.85±0.09|72.89±0.20|94.99±0.42|78.55±0.77|76.20|
> | ViT-B/16|SeMoBridge|4|71.02±0.05|91.87±0.10|91.46±0.67|34.85±0.57|62.49±0.67|76.46±2.90|71.89±0.61|86.70±0.10|72.83±0.18|95.19±0.24|78.73±0.35|75.77|
> | ViT-B/16|SeMoBridge-T (ImageNet-Trained)|4|72.04±0.04|92.60±0.44|93.91±0.23|35.67±0.59|64.60±0.78|77.94±1.53|75.39±0.43|86.67±0.08|72.87±0.16|95.20±0.28|80.78±0.30|77.06|
> | ViT-B/16|SeMoBridge-T|4|72.17±0.07|93.04±0.28|94.60±0.25|38.35±0.48|65.74±0.97|81.66±1.00|76.61±0.32|87.00±0.07|74.47±0.23|95.50±0.17|81.12±0.31|78.21|
> | ViT-B/16|SeMoBridge|8|71.53±0.05|92.02±0.09|94.14±0.64|36.81±0.55|65.59±0.31|76.48±0.81|73.42±0.14|86.85±0.11|74.03±0.15|95.60±0.42|80.58±0.38|77.00|
> | ViT-B/16|SeMoBridge-T (ImageNet-Trained)|8|73.11±0.10|93.21±0.43|96.02±0.50|39.24±0.52|66.65±1.00|76.88±1.98|77.03±0.88|87.06±0.04|74.58±0.35|95.86±0.12|82.48±0.80|78.37|
> | ViT-B/16|SeMoBridge-T|8|73.07±0.08|93.06±0.33|96.29±0.24|42.60±0.59|69.40±0.18|84.29±1.16|80.03±0.59|87.32±0.18|76.15±0.18|95.83±0.29|83.08±0.70|80.10|
> | ViT-B/16|SeMoBridge|16|71.86±0.09|92.04±0.19|95.22±0.14|39.18±0.47|66.27±0.94|78.60±0.45|76.33±0.13|86.96±0.07|74.66±0.17|95.85±0.08|81.99±0.35|78.09|
> | ViT-B/16|SeMoBridge-T (ImageNet-Trained)|16|73.96±0.22|92.75±0.49|95.74±0.37|40.33±0.38|68.74±1.47|79.56±1.34|79.61±0.46|87.05±0.08|75.28±0.34|96.02±0.20|83.15±0.29|79.29|
> | ViT-B/16|SeMoBridge-T|16|73.98±0.05|93.42±0.44|97.27±0.45|47.84±0.63|73.01±0.15|89.25±0.25|83.75±0.33|87.52±0.08|76.96±0.12|96.26±0.09|84.93±0.35|82.20|
>
> As shown in the Table, **fine-tuning on ImageNet also improves performance for different domains**.
>
> This confirms that **our method effectively learns general semantic alignment, reinforcing the claim that SeMoBridge successfully resolves intra-modal misalignment across different domains.** It makes it a highly practical "plug-and-play" module that can be deployed across various tasks without dataset-specific re-training.

---

> ### Author Response · Authors · 2025-11-21
>
> ## [W2] Dependence on CLIP
> Thank you for the comment. We acknowledge that our method is heavily reliant on CLIP. However, we emphasize that this is a strategic design choice rather than a limitation.
>
> **Our primary objective is to unlock the capabilities of powerful, pre-trained vision-language models without the high cost of re-training.**   Since CLIP is currently the standard foundation VLM, building on it maximizes practical impact that all researchers can benefit from.
>
> Furthermore, **SeMoBridge’s core principle of mapping one modality to another is not specific to CLIP**.
> It is a model-agnostic approach that can be applied to other multimodal models such as SLIP, OpenCLIP, AudioCLIP, etc.
>
> **We ran SeMoBridge on SLIP, proving that it is applicable beyond CLIP**:
>
> | **Method** | **Shots** | **OxfordPets** | **Flowers102** | **FGVCAircraft** | **DTD** | **EuroSAT** | **StanfordCars** | **Food101** | **SUN397** | **Caltech101** | **UCF101** | **ImageNet** | **AVERAGE** |
> |---|---:|---:|---:|---:|---:|---:|---:|---:|---:|---:|---:|---:|---:|
> | *SeMoBridge (SLIP ViT-B/16)* | 1 | *45.13±0.86* | *80.97±1.13* | *14.61±0.32* | *48.35±1.52* | *61.64±3.73* | *12.78±0.16* | *66.78±0.09* | *61.76±0.11* | *86.61±0.58* | *54.89±0.86* | *50.44±0.08* | 53.09 |
> | *SeMoBridge-T (SLIP ViT-B/16)* | 1 | *47.64±0.36* | *83.78±0.53* | *16.43±0.49* | *49.41±1.10* | *66.10±6.45* | *15.25±0.14* | *67.81±0.28* | *63.13±0.44* | *87.75±0.46* | *58.45±0.84* | *51.67±0.05* | *55.22* |
> | *SeMoBridge (SLIP ViT-B/16)* | 2 | *49.43±0.05* | *87.86±0.67* | *17.26±0.97* | *53.66±1.06* | *63.86±3.73* | *15.06±0.32* | *67.45±0.13* | *63.77±0.25* | *88.26±0.74* | *61.93±0.64* | *51.48±0.26* | 56.37 |
> | *SeMoBridge-T (SLIP ViT-B/16)* | 2 | *53.04±1.57* | *89.48±0.68* | *19.99±0.73* | *53.70±1.35* | *73.07±1.89* | *19.47±0.08* | *68.51±0.10* | *65.87±0.29* | *89.21±0.32* | *64.15±0.40* | *53.30±0.18* | *59.07* |
> | *SeMoBridge (SLIP ViT-B/16)* | 4 | *53.26±0.56* | *92.89±0.63* | *18.01±0.48* | *57.72±0.88* | *76.16±1.00* | *18.09±0.16* | *68.16±0.10* | *66.90±0.02* | *89.06±0.20* | *67.01±0.56* | *53.14±0.01* | 60.04 |
> | *SeMoBridge-T (SLIP ViT-B/16)* | 4 | *56.72±1.09* | *93.11±0.15* | *22.05±1.25* | *59.56±0.96* | *79.49±1.63* | *24.44±0.37* | *69.80±0.03* | *69.13±0.24* | *90.33±0.14* | *69.59±0.80* | *55.23±0.03* | *62.68* |
> | *SeMoBridge (SLIP ViT-B/16)* | 8 | *57.44±0.52* | *95.16±0.19* | *21.81±1.10* | *62.69±0.18* | *79.50±1.37* | *22.15±0.33* | *69.73±0.17* | *69.50±0.07* | *90.40±0.14* | *71.42±0.78* | *54.89±0.28* | 63.15 |
> | *SeMoBridge-T (SLIP ViT-B/16)* | 8 | *62.02±0.56* | *95.46±0.05* | *26.96±0.30* | *63.00±0.12* | *82.94±3.00* | *30.98±0.19* | *71.88±0.26* | *71.55±0.46* | *91.46±0.31* | *73.61±0.96* | *57.63±0.20* | *66.14* |
> | *SeMoBridge (SLIP ViT-B/16)* | 16 | *61.20±0.94* | *95.81±0.42* | *25.15±0.84* | *65.62±0.85* | *81.53±0.72* | *25.65±0.19* | *70.65±0.12* | *71.33±0.06* | *91.20±0.26* | *74.69±0.74* | *56.94±0.09* | 65.43 |
> | *SeMoBridge-T (SLIP ViT-B/16)* | 16 | *65.02±0.40* | *96.57±0.27* | *34.30±0.20* | *67.81±0.77* | *88.86±0.31* | *37.74±0.36* | *73.41±0.20* | *73.64±0.17* | *92.71±0.17* | *78.20±0.48* | *60.37±0.13* | *69.88* |

---

> ### Author Response · Authors · 2025-11-21
>
> ## [W3] Preservation of Visual Details
>
> We understand your concern that mapping images to the text modality might obscure fine-grained visual details. However, **we argue that this is not the case**.
>
> **The strongest evidence for this is our performance on the Image Retrieval task (see our answer to Q3)**, where we need to pick the $k$ most similar images from a gallery. For this, we bridge the query images to the text modality and compare them with the gallery.
>
> Retrieval is highly sensitive to the preservation of distinctive features.
> **If SeMoBridge had obscured critical visual details, the model would fail to distinguish between visually similar images, leading to poor retrieval performance.**

---

> ### Author Response · Authors · 2025-11-21
>
> ## [W4] Dependence on Text description
> Thank you for asking about this. We clarify that **our method does not depend on custom prompts to function effectively**, but rather benefits from them in cases where image supervision is not enough.
>
> | **Prompt Template** | **zero-shot CLIP** | **1 shot** | **2 shots** | **4 shots** | **8 shots** | **16 shots** |
> | --- | --- | --- | --- | --- | --- | --- |
> | **"a photo of a {}"** | 63.79 | 71.73 | 74.96 | 77.42 | 79.56 | 81.90 |
> | **CLIP prompts** | 62.93 | 72.10 | 75.11 | 77.42 | 79.71 | 81.83 |
> | **CLIP ensemble** | 65.87 | 72.31 | 75.23 | 77.63 | 79.73 | 82.01 |
> | **CuPL** | 68.66 | 73.88 | 76.08 | 78.20 | 80.07 | 82.19 |
> | **CLIP ensemble + CuPL** | 68.91 | 74.01 | 76.20 | 78.21 | 80.10 | 82.20 |
>
> *Figure 6 (right) is here:* https://imgur.com/2B5Xxnj
>
> As shown in the table (**Figure 6 (right) in the paper**), we have provided an ablation study addressing this concern. **SeMoBridge-T achieves strong performance (significantly outperforming baselines) even when using simple, generic prompts like “a photo of a {}”. This proves that the quality of textual descriptions is not a bottleneck.**
>
> The performance gain with LLM-generated prompts (e.g., CuPL (Pratt et al., 2023)) demonstrates our method's extensibility, not a dependency.
> By leveraging the richer textual information provided by these prompts, SeMoBridge performs a more precise transformation of image embeddings into the text modality, especially when there are very few images to work with (1-2 shots). **When more images are available (4-16 shots), the difference between generic and LLM-generated texts becomes very small.** Our method is capable of fully exploiting available semantic information from both image and text to maximize performance, while remaining effective with generic prompts.

---

> ### Author Response · Authors · 2025-11-21
>
> ## [Q1] Zero-Text Scenario
> To answer this interesting question, **we ran SeMoBridge without CLIP’s zero-shot logits ($\mathbf z_1$) and our text alignment losses ($\mathcal{L}\_{\mathrm{txte}}, \mathcal{L}\_{\mathrm{txtp}}$), which represents a scenario where no text prompts and class names are available at all.**
> This is challenging because SeMoBridge has to function with image data only.
>
> | **Method** | **IMAGENET 16 SHOT** |
> |------------|------------------------|
> | **CLIP zero-shot** | 68.73% |
> | **SeMoBridge-T (no text data)** | 68.64% |
> | **SeMoBridge-T** | 73.98% |
>
> Even when given no text data at all, SeMoBridge is able to use the 16 images per class effectively for classification and matches CLIP’s zero-shot performance.

---

> ### Author Response · Authors · 2025-11-21
>
> ## [Q2] Strategy for noisy class descriptions
> Thank you for your question. **SeMoBridge’s novelty is its multi-modal supervision strategy**, which aligns bridged embeddings with both text and image semantics. Therefore, **even if the text is noisy or ambiguous, images provide enough guidance**.
>
> An approach for mitigating issues from imperfect text prompts would be to **weight the image loss higher than the text losses**, in order to give more attention to the image semantics.

---

> ### Author Response · Authors · 2025-11-21
>
> ## [Q3] Extension to other tasks & Visual Details
> We appreciate this valuable suggestion. We agree that demonstrating performance on standard downstream tasks beyond classification significantly strengthens the paper.
>
> To validate the versatility of SeMoBridge, **we conducted additional experiments on Retrieval (both Image-Image and Text-Text)**, following the standard evaluation protocols used in related works like Cross the Gap (Mistretta et al., 2025).
>
>
> **Image-Image Retrieval (Bridging Image to Text modality):** We applied SeMoBridge to project query images into the text modality for inter-modal retrieval (picking the most similar $k$ images from a gallery).
>
> | **Model** | **Method** | **OxfordPets** | **Flowers102** | **FGVCAircraft** | **DTD** | **EuroSAT** | **StanfordCars** | **Food101** | **SUN397** | **Caltech101** | **UCF101** | **ImageNet** | **AVERAGE** |
> | --- | --- | --- | --- | --- | --- | --- | --- | --- | --- | --- | --- | --- | --- |
> | ViT-B/16 | CLIP intra-modal | 36.27 | 70.81 | 19.04 | 30.69 | **51.22** | 31.00 | — | 35.88 | 80.83 | 49.83 | — | 45.06 |
> | ViT-B/16 | SeMoBridge-T (Fast Update) | **36.96** | **74.30** | **19.54** | **34.48** | 51.21 | **34.35** | — | **37.70** | **82.78** | **52.54** | — | **47.10** |
>
>
> (Food101 and ImageNet results are missing due to VRAM issues.)
>
> **Text-Text Retrieval (Bridging Text to Image modality):** We also experimented with a **Text-to-Image version of SeMoBridge**, to map text queries into the image modality to retrieve similar texts.
> | **Model** | **Method** | **IMDB_Reviews** | **20News.** | **Climate** | **DBPedia** | **FEVER** | **NFCorpus** | **NQ** | **SciDocs** | **SciFact** | **AVERAGE** |
> | --- | --- | --- | --- | --- | --- | --- | --- | --- | --- | --- | --- |
> | ViT-B/32 | CLIP intra-modal | 52.22 | 19.24 | 11.19 | 30.32 | 58.44 | 8.90 | 23.31 | 13.54 | 26.25 | 27.05 |
> | ViT-B/32 | Cross The Gap (OVI) | 52.30 | 33.10 | 15.30 | 39.10 | 70.50 | 12.20 | 33.60 | 16.80 | 33.20 | 34.01 |
> | ViT-B/32 | SeMoBridge Text2Img | 52.81 | 41.99 | **23.38** | **43.82** | 75.78 | 13.19 | 37.95 | 18.09 | 37.99 | 38.33 |
> | ViT-B/32 | SeMoBridge-T (Fast Update) Text2Img | **57.42** | **47.99** | 21.11 | 43.58 | **76.56** | **14.02** | **41.05** | **19.33** | **40.98** | **40.23** |
>
>
> As shown in the results, SeMoBridge consistently improves performance in both Image-to-Text and Text-to-Image retrieval tasks. Crucially, **it confirms that our approach addresses intra-modal misalignment effectively** and translates it into improvements in many standard downstream tasks.
>
> **Regarding the concern about visual details, we refer the reviewer to our response in [W3]**.
>
> By utilizing CLIP’s calibrated inter-modal space for same-modality comparisons, **SeMoBridge effectively removes intra-modal misalignment while preserving the discriminative visual attributes** required for detail-sensitive tasks.

---

### Official Review · Reviewer_hnwV · 2025-11-01

**Soundness:** 3
**Presentation:** 3
**Contribution:** 3
**Rating:** 6
**Confidence:** 1

**Summary:**

CLIP variants and few-shot image classification are out of my expertise, and I write these review comments based on my general knowledge and understanding.

This paper focuses on the intra-modal calibration problem, where images with different classes may introduce confusions. To address this problem, the authors propose SeMoBridge, which integrates the fine-grained text modality to optimize image representations. Therefore the performance on few-shot image classification is improved. The overall structure is lightweight and very easy to train, and it also supports a training-free setting.

**Strengths:**

- The training paradigm is very efficient and lightweight, and only a projector/adapter is needed. Approximately less than 1M parameters are required and the whole training process requires less than one minute.
- The proposed method is effective and outperforms other baselines. Besides, SeMoBridge is better in fewer shots.
- The writing and overall presentation is good, and the structure is well organized.
- The experiments are solid and 11 datasets are used for evaluation.

**Weaknesses:**

- More shots lead to smaller performance gap compared to the baselines in Figure 4, 5, and Figure 6 (right). Further explanations should be provided.
- It seems SeMoBridge could be a plugin to any CLIP-related tasks, but current paper does not extend it to further domains and tasks.
- Besides Figure 2 right and Figure 7, there lacks of a quantitative metric to explicitly express the levels of intra-modal calibration problem. Currently, all the results are based on general final performance.

**Questions:**

- Does few-shot text classification also has such a intra-modality calibration problem, and can we map these texts to images for effective calibration?

---

> ### Author Response · Authors · 2025-11-21
>
> We thank the reviewer for their time and their helpful feedback on our submission. We believe that it provides valuable perspectives that will enhance the quality and clarity of our work.
>
> **Below, we provide detailed responses addressing each of the weaknesses [W#] and questions [Q#] raised.**
>
> ---
>
> ## [W1] Performance across shot-settings
> Thank you for asking about this. **We believe this phenomenon actually highlights the core strength and intended contribution of our method.**
>
> As the number of shots increases, the few-shot set naturally provides a better statistical representation of the class distribution. Consequently, most baseline methods eventually acquire sufficient information to perform reasonably well, leading to a natural "saturation" where the performance gap between methods narrows.
>
> **The true challenge lies in the low-shot settings (1–4 shots) where visual information is scarce.**
> In these scenarios, other models struggle to extract reliable features from images. This is exactly where SeMoBridge-T demonstrates a unique advantage. **Our method improves the 1-shot SOTA (CLIP-LoRA) by 1.25% while reducing training time by a factor of 28. We beat the previous fastest method, LDC (Li et al., 2025), by 2.26% while requiring only ¼ of the training time.**
>
> Unlike baselines that may not fully utilize auxiliary textual knowledge, SeMoBridge's training is specifically designed to align bridged visual features with detailed semantic descriptions. **We have also discussed this mechanism and the impact of such multi-modal supervision in the "Impact of loss terms" section on page 8 of our submission.**
>
> Therefore, **the larger performance gap in lower shots is strong evidence that SeMoBridge successfully maximizes the utility of textual semantics when visual data is insufficient**.

---

> ### Author Response · Authors · 2025-11-21
>
> ## [W2] Generalizability beyond Classification
>
> Thank you for recognizing the potential of SeMoBridge as a general plugin. We agree that demonstrating performance beyond classification strengthens our contribution.
>
> To validate the versatility of SeMoBridge, **we conducted additional experiments on Retrieval (both Image-Image and Text-Text)**, following the standard evaluation protocols used in related works like Cross the Gap (Mistretta et al., 2025).
>
> **Image-Image Retrieval (Bridging Image to Text modality):** We applied SeMoBridge to project query images into the text modality for inter-modal retrieval (picking the most similar $k$ images from a gallery).
>
> | **Model** | **Method** | **OxfordPets** | **Flowers102** | **FGVCAircraft** | **DTD** | **EuroSAT** | **StanfordCars** | **Food101** | **SUN397** | **Caltech101** | **UCF101** | **ImageNet** | **AVERAGE** |
> |---|---|---|---|---|---|---|---|---|---|---|---|---|---|
> | ViT-B/16 | CLIP intra-modal | 36.27 | 70.81 | 19.04 | 30.69 | **51.22** | 31.00 | — | 35.88 | 80.83 | 49.83 | — | 45.06 |
> | ViT-B/16 | SeMoBridge-T (Fast Update) | **36.96** | **74.30** | **19.54** | **34.48** | 51.21 | **34.35** | — | **37.70** | **82.78** | **52.54** | — | **47.10** |
>
> (Food101 and ImageNet results are missing due to VRAM issues.)
>
> In the answer to Q1, we provide Text-Retrieval results.
>
> As shown, SeMoBridge consistently improves performance in both Image-to-Text and Text-to-Image retrieval tasks. Crucially, **it confirms that our approach addresses intra-modal misalignment effectively** and translates it into improvements in many standard downstream tasks.
>
> Additionally, **we explored SeMoBridge’s generalizability by fine-tuning the bridge on ImageNet’s few-shot set (1-16 shots), and then evaluating it on other datasets (dataset transfer).**
>
> |Model|Method|Shots|ImageNet|OxfordPets|Flowers102|FGVCAircraft|DTD|EuroSAT|StanfordCars|Food101|SUN397|Caltech101|UCF101|AVERAGE|
> |---|---|---|---|---|---|---|---|---|---|---|---|---|---|---|
> |ViT-B/16|CLIP zero-shot|0|68.73±0.00|89.10±0.00|70.73±0.00|24.69±0.00|44.09±0.00|48.31±0.00|65.61±0.00|85.87±0.00|62.59±0.00|93.35±0.00|67.62±0.00|65.52|
> | ViT-B/16|SeMoBridge|1|70.28±0.05|91.79±0.19|84.06±0.64|30.40±0.92|56.89±0.69|68.18±4.30|68.15±0.66|86.42±0.03|70.24±0.34|94.36±0.28|73.95±1.24|72.25|
> | ViT-B/16|SeMoBridge-T (ImageNet-Trained)|1|70.67±0.03|92.12±0.10|85.09±0.70|29.60±0.47|55.52±0.64|66.78±0.92|68.15±0.62|86.47±0.06|70.56±0.37|94.55±0.20|72.39±0.37|71.99|
> | ViT-B/16|SeMoBridge-T|1|70.88±0.09|92.22±0.19|89.84±0.85|32.94±0.43|59.79±0.64|69.69±5.48|70.27±0.44|86.62±0.04|71.17±0.21|94.85±0.20|75.87±0.56|74.01|
> | ViT-B/16|SeMoBridge|2|70.66±0.11|91.61±0.49|89.21±0.50|32.34±0.26|59.47±1.10|70.67±0.89|69.88±0.15|86.55±0.01|71.19±0.05|94.85±0.20|76.60±1.40|73.91|
> | ViT-B/16|SeMoBridge-T (ImageNet-Trained)|2|71.28±0.12|92.22±0.18|90.58±0.75|32.42±0.22|59.14±0.62|68.99±2.00|71.78±0.97|86.57±0.03|71.59±0.25|94.86±0.27|76.91±0.67|74.21|
> | ViT-B/16|SeMoBridge-T|2|71.40±0.04|92.24±0.22|92.03±0.65|35.28±0.71|61.90±1.09|78.65±2.96|73.46±0.75|86.85±0.09|72.89±0.20|94.99±0.42|78.55±0.77|76.20|
> | ViT-B/16|SeMoBridge|4|71.02±0.05|91.87±0.10|91.46±0.67|34.85±0.57|62.49±0.67|76.46±2.90|71.89±0.61|86.70±0.10|72.83±0.18|95.19±0.24|78.73±0.35|75.77|
> | ViT-B/16|SeMoBridge-T (ImageNet-Trained)|4|72.04±0.04|92.60±0.44|93.91±0.23|35.67±0.59|64.60±0.78|77.94±1.53|75.39±0.43|86.67±0.08|72.87±0.16|95.20±0.28|80.78±0.30|77.06|
> | ViT-B/16|SeMoBridge-T|4|72.17±0.07|93.04±0.28|94.60±0.25|38.35±0.48|65.74±0.97|81.66±1.00|76.61±0.32|87.00±0.07|74.47±0.23|95.50±0.17|81.12±0.31|78.21|
> | ViT-B/16|SeMoBridge|8|71.53±0.05|92.02±0.09|94.14±0.64|36.81±0.55|65.59±0.31|76.48±0.81|73.42±0.14|86.85±0.11|74.03±0.15|95.60±0.42|80.58±0.38|77.00|
> | ViT-B/16|SeMoBridge-T (ImageNet-Trained)|8|73.11±0.10|93.21±0.43|96.02±0.50|39.24±0.52|66.65±1.00|76.88±1.98|77.03±0.88|87.06±0.04|74.58±0.35|95.86±0.12|82.48±0.80|78.37|
> | ViT-B/16|SeMoBridge-T|8|73.07±0.08|93.06±0.33|96.29±0.24|42.60±0.59|69.40±0.18|84.29±1.16|80.03±0.59|87.32±0.18|76.15±0.18|95.83±0.29|83.08±0.70|80.10|
> | ViT-B/16|SeMoBridge|16|71.86±0.09|92.04±0.19|95.22±0.14|39.18±0.47|66.27±0.94|78.60±0.45|76.33±0.13|86.96±0.07|74.66±0.17|95.85±0.08|81.99±0.35|78.09|
> | ViT-B/16|SeMoBridge-T (ImageNet-Trained)|16|73.96±0.22|92.75±0.49|95.74±0.37|40.33±0.38|68.74±1.47|79.56±1.34|79.61±0.46|87.05±0.08|75.28±0.34|96.02±0.20|83.15±0.29|79.29|
> | ViT-B/16|SeMoBridge-T|16|73.98±0.05|93.42±0.44|97.27±0.45|47.84±0.63|73.01±0.15|89.25±0.25|83.75±0.33|87.52±0.08|76.96±0.12|96.26±0.09|84.93±0.35|82.20|
>
> As shown in the Table, **fine-tuning on ImageNet also improves performance for different domains**.
>
> This confirms that **our method effectively learns general semantic alignment, reinforcing the claim that SeMoBridge successfully resolves intra-modal misalignment across different domains.** It makes it a highly practical "plug-and-play" module that can be deployed across various tasks without dataset-specific re-training.

---

> ### Author Response · Authors · 2025-11-21
>
> ## [W3] Quantitative proof of misalignment reduction
> Thank you for pointing this out.
> **To quantitatively showcase how well SeMoBridge addresses intra-modal misalignment, we calculated the “overlap” (or confusion) between unpaired and paired image cosine similarity distributions.**
>
>
> *A figure showing the cosine similarity distribution overlap:* https://imgur.com/XrbD97g
>
> | **Comparison Method** | **Unpaired/Paired overlap** |
> |---|---|
> | CLIP Image-Text | 0.71% |
> | Intra-Modal Image-Image | 22.99% |
> | SeMoBridge Image-Image | 0.78% |
> | SeMoBridge-T Image-Image | **0.02%** |
>
> As shown, it is highly reduced after enabling inter-modal comparisons using our method (from 22.99% to 0.78% and 0.02% for SeMoBridge and SeMoBridge-T, respectively).
>
> Consequently, **these more robust comparisons improve downstream tasks by relying on well-aligned inter-modal interactions rather than noisy intra-modal ones**. This mechanism is what yields our superior results compared to other adaptation methods.
>
> We also plan to provide these metrics for each dataset in the appendix.

---

> ### Author Response · Authors · 2025-11-21
>
> ## [Q1] Application to text modality
>
> This is an excellent question that touches upon the universality of our approach.
> To investigate whether mapping between modalities improves calibration even for text, **we built a text-to-image version of SeMoBridge**.
>
> Instead of the suggested few-shot text classification task, we decided to follow Cross the Gap (Mistretta et al., 2025) for the text-retrieval task.
> **This SeMoBridge version maps text queries into the image modality** to retrieve the $k$ most similar texts from a gallery.
>
> | **Model** | **Method** | **IMDB_Reviews** | **20News.** | **Climate** | **DBPedia** | **FEVER** | **NFCorpus** | **NQ** | **SciDocs** | **SciFact** | **AVERAGE** |
> |---|---|---|---|---|---|---|---|---|---|---|---|
> | ViT-B/32 | CLIP intra-modal | 52.22 | 19.24 | 11.19 | 30.32 | 58.44 | 8.90 | 23.31 | 13.54 | 26.25 | 27.05 |
> | ViT-B/32 | Cross The Gap (OVI) | 52.30 | 33.10 | 15.30 | 39.10 | 70.50 | 12.20 | 33.60 | 16.80 | 33.20 | 34.01 |
> | ViT-B/32 | SeMoBridge Text2Img | 52.81 | 41.99 | **23.38** | **43.82** | 75.78 | 13.19 | 37.95 | 18.09 | 37.99 | 38.33 |
> | ViT-B/32 | SeMoBridge-T (Fast Update) Text2Img | **57.42** | **47.99** | 21.11 | 43.58 | **76.56** | **14.02** | **41.05** | **19.33** | **40.98** | **40.23** |
>
> As shown in the table, SeMoBridge consistently improves performance in text retrieval tasks.
> **It confirms that both modalities have this problem, and that our approach also works for the text modality.**

---

> > ### Comment · Reviewer_hnwV · 2025-11-25
> >
> > Thanks for the thorough explanations and the new experimental results. I think you have addressed all my concerns. I'd like to hearing from the othe reviewers before making my final assessment because I'm not familiar with this subject and will maintain my low confidence score.

---

### Official Review · Reviewer_JR1v · 2025-11-03

**Soundness:** 2
**Presentation:** 2
**Contribution:** 1
**Rating:** 2
**Confidence:** 4

**Summary:**

This paper considers the problem of the modality gap in CLIP's vision–language modeling, which leads to an embedding space with potentiallhy undesirable properties and arises from its inter-modal contrastive training. The paper focuses in particular on undesirable intra-modality comparisons in the visual embeddings.

To address these, the paper proposes a simple mapping of visual embeddings into the textual embedding space, which is particularly fast to obtain, as it can be done in a training-free setup or with quick fine-tuning.

**Strengths:**

- The approach is super-fast.

**Weaknesses:**

- The paper proposes a simple technique to obtain embeddings of images in the text embedding space and claim that this addresses the modality gap. While it is true that the resulting embeddings reside in the same space, it was always clear that you can achieve such a projection to have embeddings in the same space. However, genuinely solving CLIP's modality gap refers to something else: It involves developing an improved learning paradigm that leads to embeddings that enable us to do everything we can normally do with CLIP plus also have an embedding space without any modality gap. The submission investigates a single use case (image few-shot classification) without considering the other requirements.

- While the modality gap is well-established, the motivation in Figure 1 seems more abstract: The left side shows an idealized example where a dog image is closer to a cat centroid. However, it is  unclear whether such a case (with these specific example images) would ever occur with CLIP. I imagine this would happen more with easier to confuse classes. The right side of Figure 1 appears to be a more data-driven analysis, but again lacks crucial details for reproducibility, e.g. the rows and columns are not labeled with any classes, so it is hard to know in what instances this problem might occur.

- The approach is directly based on SD-IPC (Ding et al., 2023), a method to create a textual embedding for images. The authors adapt this approach for CLIP-based few-shot classification.

- The success of the approach hinges a lot on the quality of the textual descriptions. The paper uses custom prompts for each dataset to obtain appropriate class descriptions instead of a generic solution.

**Questions:**

- Can you report additional experiments on standard CLIP downstream tasks?

- Is there a way to avoid the use of class-specific bias (CSB) vectors? They appear to severely limit the generalizability of the approach.

---

> ### Author Response · Authors · 2025-11-21
>
> We thank the reviewer for their time and their helpful feedback on our submission. We believe that it provides valuable perspectives that will enhance the quality and clarity of our work.
>
> **Below, we provide detailed responses addressing each of the weaknesses [W#] and questions [Q#] raised.**
>
> ---
>
> ## [W1] The goal of SeMoBridge
> We agree that developing a new pre-training paradigm to fundamentally eliminate the modality gap is a grand and valuable goal. However, **we respectfully argue that addressing the “intra-modal misalignment” (which is caused by the modality gap) in a post-hoc manner is an equally critical research contribution**, and here is why:
>
> **While re-training CLIP might theoretically solve the gap, it requires massive computational resources and data, making it inaccessible for most researchers and practitioners.** Our method, SeMoBridge, “unlocks” the potential of frozen, pre-trained models for more tasks without the need for expensive re-training.
>
> **To quantitatively showcase how well SeMoBridge addresses intra-modal misalignment, we calculated the “overlap” (or confusion) between unpaired and paired image cosine similarity distributions.**
>
> *A figure showing the cosine similarity distribution overlap:* https://imgur.com/XrbD97g
>
> | **Comparison Method** | **Unpaired/Paired overlap** |
> | --- | --- |
> | CLIP Image-Text | 0.71% |
> | Intra-Modal Image-Image | 22.99% |
> | SeMoBridge Image-Image | 0.78% |
> | SeMoBridge-T Image-Image | **0.02%** |
>
> As shown, it is highly reduced after enabling inter-modal comparisons using our method (from 22.99% to 0.78% and 0.02% for SeMoBridge and SeMoBridge-T, respectively).
>
> Consequently, **these more robust comparisons improve downstream tasks by relying on well-aligned inter-modal interactions rather than noisy intra-modal ones**. This mechanism is what yields our superior results compared to other adaptation methods.
>
> However, we also agree that exploring additional tasks with SeMoBridge will significantly enhance the paper’s contributions. For this reason, **we provide image-retrieval and text-retrieval (with text-to-image SeMoBridge) benchmarks in the answer to Q1.**
>
> We also plan to provide these metrics for each dataset in the appendix.

---

> ### Author Response · Authors · 2025-11-21
>
> ## [W2] Improvements on Figure 2
> We appreciate your detailed analysis of our motivational figure, and fully agree with your constructive feedback that the analysis lacked specific details. To improve reproducibility and clarity, we have added class labels to the rows and columns of the confusion matrix.
>
> *New Figure 2*: https://imgur.com/QqoxfZX
>
> **We chose the dog/cat example for conceptual clarity to intuitively show the misalignment problem.** This choice also mirrors the motivational examples used in prior foundational work, such as Cross the Gap (Mistretta et al., 2025).
>
>
>
> *Cross the Gap's similar figure:* https://imgur.com/ndk8IjI
>
> *Dog/Cat image similarity distributions:* https://imgur.com/8yh4ek9
>
> As you rightly questioned, one might assume these classes are easy to distinguish. However, as detailed in Appendix B of Mistretta et al. (2025), the degree of class confusion (misalignment) between even seemingly distinct classes like dogs and cats is significant in CLIP's embedding space. Thus, this is not an idealized case but a reflection of an underlying issue.

---

> ### Author Response · Authors · 2025-11-21
>
> ## [W3] Our novelty over SD-IPC
>
> SD-IPC operates in the generative domain (Stable Diffusion) to invert images into text tokens for image generation. In contrast, **we identify for the first time that this mathematical intuition can be repurposed to solve tasks like intra-modal misalignment in classification and retrieval**. Transferring insights across such distinct domains is a non-trivial contribution.
>
> We argue that this undermines our key contribution. Unlike SD-IPC, **SeMoBridge-T introduces a novel multi-modal supervised training strategy**. This includes loss formulations to improve the bridge’s semantic alignment ($\mathcal L_\textrm{img}$, $\mathcal L_\textrm{txte}$, $\mathcal L_\textrm{txtp}$) and the Class-Specific Bias (CSB) term designed for classification boundaries. These components are absent in SD-IPC, proving that our method goes beyond simple application.

---

> ### Author Response · Authors · 2025-11-21
>
> ## [W4] Dependency on Custom Prompts
> Thank you for asking about this. We clarify that **our method does not depend on custom prompts to function effectively**, but rather benefits from them in cases where image supervision is not enough.
>
> | **Prompt Template** | **zero-shot CLIP** | **1 shot** | **2 shots** | **4 shots** | **8 shots** | **16 shots** |
> | --- | --- | --- | --- | --- | --- | --- |
> | **"a photo of a {}"** | 63.79 | 71.73 | 74.96 | 77.42 | 79.56 | 81.90 |
> | **CLIP prompts** | 62.93 | 72.10 | 75.11 | 77.42 | 79.71 | 81.83 |
> | **CLIP ensemble** | 65.87 | 72.31 | 75.23 | 77.63 | 79.73 | 82.01 |
> | **CuPL** | 68.66 | 73.88 | 76.08 | 78.20 | 80.07 | 82.19 |
> | **CLIP ensemble + CuPL** | 68.91 | 74.01 | 76.20 | 78.21 | 80.10 | 82.20 |
>
> *Figure 6 (right) is here:* https://imgur.com/2B5Xxnj
>
> As shown in the table (**Figure 6 (right) in the paper**), we have provided an ablation study addressing this concern. SeMoBridge-T achieves strong performance (significantly outperforming baselines) even when using simple, generic prompts like “a photo of a {}”. This proves that the quality of textual descriptions is not a bottleneck.
>
> The performance gain with LLM-generated prompts (e.g., CuPL (Pratt et al., 2023)) demonstrates our method's extensibility, not a dependency.
> By leveraging the richer textual information provided by these prompts, SeMoBridge performs a more precise transformation of image embeddings into the text modality, especially when there are very few images to work with (1-2 shots). **When more images are available (4-16 shots), the difference between generic and LLM-generated texts becomes very small.** Our method is capable of fully exploiting available semantic information from both image and text to maximize performance, while remaining effective with generic prompts.
>
> **Using CuPL prompts is not specific to us**; it follows the standard setting of recent state-of-the-art methods (e.g., APE (Zhu et al., 2023)) to ensure fair comparison.

---

> ### Author Response · Authors · 2025-11-21
>
> ## [Q1] Downstream Tasks: Retrieval
> We appreciate this valuable suggestion. We agree that demonstrating performance on standard downstream tasks beyond classification significantly strengthens the paper.
>
> To validate the versatility of SeMoBridge, **we conducted additional experiments on Retrieval (both Image-Image and Text-Text)**, following the standard evaluation setting used in related works like Cross the Gap (Mistretta et al., 2025).
>
> **Image-Image Retrieval (Bridging Image to Text modality):** We applied SeMoBridge to project query images into the text modality for inter-modal retrieval (picking the most similar $k$ images from a gallery).
>
> | **Model** | **Method** | **OxfordPets** | **Flowers102** | **FGVCAircraft** | **DTD** | **EuroSAT** | **StanfordCars** | **Food101** | **SUN397** | **Caltech101** | **UCF101** | **ImageNet** | **AVERAGE** |
> | --- | --- | --- | --- | --- | --- | --- | --- | --- | --- | --- | --- | --- | --- |
> | ViT-B/16 | CLIP intra-modal | 36.27 | 70.81 | 19.04 | 30.69 | **51.22** | 31.00 | — | 35.88 | 80.83 | 49.83 | — | 45.06 |
> | ViT-B/16 | SeMoBridge-T (Fast Update) | **36.96** | **74.30** | **19.54** | **34.48** | 51.21 | **34.35** | — | **37.70** | **82.78** | **52.54** | — | **47.10** |
>
> (Food101 and ImageNet results are missing due to VRAM issues.)
>
>
> **Text-Text Retrieval (Bridging Text to Image modality):** We also experimented with a **Text-to-Image version of SeMoBridge**, to map text queries into the image modality to retrieve similar texts.
> | **Model** | **Method** | **IMDB_Reviews** | **20News.** | **Climate** | **DBPedia** | **FEVER** | **NFCorpus** | **NQ** | **SciDocs** | **SciFact** | **AVERAGE** |
> | --- | --- | --- | --- | --- | --- | --- | --- | --- | --- | --- | --- |
> | ViT-B/32 | CLIP intra-modal | 52.22 | 19.24 | 11.19 | 30.32 | 58.44 | 8.90 | 23.31 | 13.54 | 26.25 | 27.05 |
> | ViT-B/32 | Cross The Gap (OVI) | 52.30 | 33.10 | 15.30 | 39.10 | 70.50 | 12.20 | 33.60 | 16.80 | 33.20 | 34.01 |
> | ViT-B/32 | SeMoBridge Text2Img | 52.81 | 41.99 | **23.38** | **43.82** | 75.78 | 13.19 | 37.95 | 18.09 | 37.99 | 38.33 |
> | ViT-B/32 | SeMoBridge-T (Fast Update) Text2Img | **57.42** | **47.99** | 21.11 | 43.58 | **76.56** | **14.02** | **41.05** | **19.33** | **40.98** | **40.23** |
>
>
> As shown in the results, SeMoBridge consistently improves performance in both Image-to-Text and Text-to-Image retrieval tasks. Crucially, **it confirms that our approach addresses intra-modal misalignment effectively** and translates it into improvements in many standard downstream tasks.

---

> ### Author Response · Authors · 2025-11-21
>
> ## [Q2] Role of CSB & Transferability (1/2)
> Thank you for pointing this out. **The CSB is not a core component of our method and is not required for it to perform well.**
>
> |Model|Method|Shots|OxfordPets|Flowers102|FGVCAircraft|DTD|EuroSAT|StanfordCars|Food101|SUN397|Caltech101|UCF101|ImageNet|AVERAGE|
> |---|---|---|---|---|---|---|---|---|---|---|---|---|---|---|
> |ViT-B/16|CLIP zero-shot|0|89.10±0.00|70.73±0.00|24.69±0.00|44.09±0.00|48.31±0.00|65.61±0.00|85.87±0.00|62.59±0.00|93.35±0.00|67.62±0.00|68.73±0.00|**65.52**|
> |ViT-B/16|SeMoBridge|1|91.79±0.19|84.06±0.64|30.40±0.92|56.89±0.69|68.18±4.30|68.15±0.66|86.42±0.03|70.24±0.34|94.36±0.28|73.95±1.24|70.28±0.05|**72.25**|
> | ViT-B/16|SeMoBridge-T (NoCSB)|1|92.30±0.13|89.61±0.81|32.81±0.74|59.91±0.45|69.78±5.48|69.78±0.34|86.67±0.02|71.01±0.14|94.70±0.25|76.04±0.84|70.67±0.03|**73.93**|
> | ViT-B/16|SeMoBridge-T|1|92.22±0.19|89.84±0.85|32.94±0.43|59.79±0.64|69.69±5.48|70.27±0.44|86.62±0.04|71.17±0.21|94.85±0.20|75.87±0.56|70.88±0.09|**74.01**|
> | ViT-B/16|SeMoBridge|2|91.61±0.49|89.21±0.50|32.34±0.26|59.47±1.10|70.67±0.89|69.88±0.15|86.55±0.01|71.19±0.05|94.85±0.20|76.60±1.40|70.66±0.11|**73.91**|
> | ViT-B/16|SeMoBridge-T (NoCSB)|2|92.22±0.22|92.18±0.74|35.42±0.48|61.82±1.47|78.69±2.85|73.66±0.23|86.85±0.06|72.36±0.12|94.74±0.43|78.25±0.91|71.28±0.12|**76.13**|
> | ViT-B/16|SeMoBridge-T|2|92.24±0.22|92.03±0.65|35.28±0.71|61.90±1.09|78.65±2.96|73.46±0.75|86.85±0.09|72.89±0.20|94.99±0.42|78.55±0.77|71.40±0.04|**76.20**|
> | ViT-B/16|SeMoBridge|4|91.87±0.10|91.46±0.67|34.85±0.57|62.49±0.67|76.46±2.90|71.89±0.61|86.70±0.10|72.83±0.18|95.19±0.24|78.73±0.35|71.02±0.05|**75.77**|
> | ViT-B/16|SeMoBridge-T (NoCSB)|4|92.57±0.19|94.70±0.28|38.80±0.29|65.70±0.95|81.80±1.34|77.02±0.52|87.00±0.04|74.27±0.28|95.42±0.09|81.10±0.42|72.04±0.04|**78.22**|
> | ViT-B/16|SeMoBridge-T|4|93.04±0.28|94.60±0.25|38.35±0.48|65.74±0.97|81.66±1.00|76.61±0.32|87.00±0.07|74.47±0.23|95.50±0.17|81.12±0.31|72.17±0.07|**78.21**|
> | ViT-B/16|SeMoBridge|8|92.02±0.09|94.14±0.64|36.81±0.55|65.59±0.31|76.48±0.81|73.42±0.14|86.85±0.11|74.03±0.15|95.60±0.42|80.58±0.38|71.53±0.05|**77.00**|
> | ViT-B/16|SeMoBridge-T (NoCSB)|8|92.92±0.39|96.25±0.38|43.62±0.80|69.27±0.17|84.35±1.07|80.31±0.67|87.39±0.16|75.87±0.07|95.71±0.04|83.36±0.52|73.11±0.10|**80.20**|
> | ViT-B/16|SeMoBridge-T|8|93.06±0.33|96.29±0.24|42.60±0.59|69.40±0.18|84.29±1.16|80.03±0.59|87.32±0.18|76.15±0.18|95.83±0.29|83.08±0.70|73.07±0.08|**80.10**|
> | ViT-B/16|SeMoBridge|16|92.04±0.19|95.22±0.14|39.18±0.47|66.27±0.94|78.60±0.45|76.33±0.13|86.96±0.07|74.66±0.17|95.85±0.08|81.99±0.35|71.86±0.09|**78.09**|
> | ViT-B/16|SeMoBridge-T (NoCSB)|16|93.58±0.16|96.94±0.14|48.61±0.54|72.78±0.56|89.37±0.36|83.85±0.45|87.58±0.07|77.14±0.06|96.31±0.09|85.07±0.13|73.96±0.22|**82.29**|
> | ViT-B/16|SeMoBridge-T|16|93.42±0.44|97.27±0.45|47.84±0.63|73.01±0.15|89.25±0.25|83.75±0.33|87.52±0.08|76.96±0.12|96.26±0.09|84.93±0.35|73.98±0.05|**82.20**|
>
>
> As shown in the Table, SeMoBridge-T achieves very similar accuracy with or without the CSB. We included this optional term to show a slightly more stabilized version for cases that might require it, but the core approach is effective and generalizable on its own. We provide a full analysis of CSB in Appendix A.3.

---

> ### Author Response · Authors · 2025-11-21
>
> ## [Q2] Role of CSB & Transferability (2/2)
>
> Additionally, **we explored SeMoBridge’s generalizability further by fine-tuning the bridge on ImageNet’s few-shot set (1-16 shots), and then evaluating it on other datasets (dataset transfer).**
>
> |Model|Method|Shots|ImageNet|OxfordPets|Flowers102|FGVCAircraft|DTD|EuroSAT|StanfordCars|Food101|SUN397|Caltech101|UCF101|AVERAGE|
> |---|---|---|---|---|---|---|---|---|---|---|---|---|---|---|
> |ViT-B/16|CLIP zero-shot|0|68.73±0.00|89.10±0.00|70.73±0.00|24.69±0.00|44.09±0.00|48.31±0.00|65.61±0.00|85.87±0.00|62.59±0.00|93.35±0.00|67.62±0.00|65.52|
> | ViT-B/16|SeMoBridge|1|70.28±0.05|91.79±0.19|84.06±0.64|30.40±0.92|56.89±0.69|68.18±4.30|68.15±0.66|86.42±0.03|70.24±0.34|94.36±0.28|73.95±1.24|72.25|
> | ViT-B/16|SeMoBridge-T (ImageNet-Trained)|1|70.67±0.03|92.12±0.10|85.09±0.70|29.60±0.47|55.52±0.64|66.78±0.92|68.15±0.62|86.47±0.06|70.56±0.37|94.55±0.20|72.39±0.37|71.99|
> | ViT-B/16|SeMoBridge-T|1|70.88±0.09|92.22±0.19|89.84±0.85|32.94±0.43|59.79±0.64|69.69±5.48|70.27±0.44|86.62±0.04|71.17±0.21|94.85±0.20|75.87±0.56|74.01|
> | ViT-B/16|SeMoBridge|2|70.66±0.11|91.61±0.49|89.21±0.50|32.34±0.26|59.47±1.10|70.67±0.89|69.88±0.15|86.55±0.01|71.19±0.05|94.85±0.20|76.60±1.40|73.91|
> | ViT-B/16|SeMoBridge-T (ImageNet-Trained)|2|71.28±0.12|92.22±0.18|90.58±0.75|32.42±0.22|59.14±0.62|68.99±2.00|71.78±0.97|86.57±0.03|71.59±0.25|94.86±0.27|76.91±0.67|74.21|
> | ViT-B/16|SeMoBridge-T|2|71.40±0.04|92.24±0.22|92.03±0.65|35.28±0.71|61.90±1.09|78.65±2.96|73.46±0.75|86.85±0.09|72.89±0.20|94.99±0.42|78.55±0.77|76.20|
> | ViT-B/16|SeMoBridge|4|71.02±0.05|91.87±0.10|91.46±0.67|34.85±0.57|62.49±0.67|76.46±2.90|71.89±0.61|86.70±0.10|72.83±0.18|95.19±0.24|78.73±0.35|75.77|
> | ViT-B/16|SeMoBridge-T (ImageNet-Trained)|4|72.04±0.04|92.60±0.44|93.91±0.23|35.67±0.59|64.60±0.78|77.94±1.53|75.39±0.43|86.67±0.08|72.87±0.16|95.20±0.28|80.78±0.30|77.06|
> | ViT-B/16|SeMoBridge-T|4|72.17±0.07|93.04±0.28|94.60±0.25|38.35±0.48|65.74±0.97|81.66±1.00|76.61±0.32|87.00±0.07|74.47±0.23|95.50±0.17|81.12±0.31|78.21|
> | ViT-B/16|SeMoBridge|8|71.53±0.05|92.02±0.09|94.14±0.64|36.81±0.55|65.59±0.31|76.48±0.81|73.42±0.14|86.85±0.11|74.03±0.15|95.60±0.42|80.58±0.38|77.00|
> | ViT-B/16|SeMoBridge-T (ImageNet-Trained)|8|73.11±0.10|93.21±0.43|96.02±0.50|39.24±0.52|66.65±1.00|76.88±1.98|77.03±0.88|87.06±0.04|74.58±0.35|95.86±0.12|82.48±0.80|78.37|
> | ViT-B/16|SeMoBridge-T|8|73.07±0.08|93.06±0.33|96.29±0.24|42.60±0.59|69.40±0.18|84.29±1.16|80.03±0.59|87.32±0.18|76.15±0.18|95.83±0.29|83.08±0.70|80.10|
> | ViT-B/16|SeMoBridge|16|71.86±0.09|92.04±0.19|95.22±0.14|39.18±0.47|66.27±0.94|78.60±0.45|76.33±0.13|86.96±0.07|74.66±0.17|95.85±0.08|81.99±0.35|78.09|
> | ViT-B/16|SeMoBridge-T (ImageNet-Trained)|16|73.96±0.22|92.75±0.49|95.74±0.37|40.33±0.38|68.74±1.47|79.56±1.34|79.61±0.46|87.05±0.08|75.28±0.34|96.02±0.20|83.15±0.29|79.29|
> | ViT-B/16|SeMoBridge-T|16|73.98±0.05|93.42±0.44|97.27±0.45|47.84±0.63|73.01±0.15|89.25±0.25|83.75±0.33|87.52±0.08|76.96±0.12|96.26±0.09|84.93±0.35|82.20|
>
> As shown in the Table, **fine-tuning on ImageNet also improves performance for different domains**.
>
> This confirms that **our method effectively learns general semantic alignment, reinforcing the claim that SeMoBridge successfully resolves intra-modal misalignment across different domains.** It makes it a highly practical "plug-and-play" module that can be deployed across various tasks without dataset-specific re-training.

---

### Meta-Review · Area_Chair_2y7b · 2026-01-09

**Summary:**

This paper received mixed reviews: two borderline accepts, one borderline reject, and one reject scores. The reviewers appreciated the simple, lightweight, and effective method, its clear motivation and performance gain, extensive evaluation, and clarity of the manuscript. However, they raised an unanimous concern with the narrow focus and limited evaluation settings, which look crucial. Moreover, the reviewers pointed out reliance on the custom prompts (JR1v, WGAS), incremental novelty (JR1v), lack of theoretical justification (WGAS), limited versatility of the method (ay5F), marginal performance gain under more shots (hnwV), lack of detailed analysis on the quantitative results (hnwV), and issues on conceptual illustration in Figure 2 (JR1v). The authors responded to these comments through the rebuttal and revision, but failed to assuage all of them. In particular, the AC found that the concerns with the narrow focus, limited evaluation settings, and the dependence on the custom prompts remain unresolved clearly. The authors conducted additional experiments in expanded settings and tasks, but they are still limited to retrieval tasks, and more importantly, fail to demonstrate the advantages of CLIP embeddings. Also, although the rebuttal offers new empirical analysis on the impact of text prompts, the results do not well support the authors' claim due to the performance degradation by the use of simpler text prompts; more detailed quantitative analysis would be required to assuage the concern. Putting these together, the AC found that the remaining concerns outweigh the positive comments and the rebuttal, and thus regrets to recommend rejection. The authors are encouraged to improve the paper with the valuable comments from the reviewers, and resubmit to next venues.

**Reviewer Concerns:**

[Concerns that have not been fully resolved]
- Narrow focus of the paper and limited evaluation (all)
	- *Reviewer JR1v claimed that the paper does not explore the generalization and broad applicability of the proposed method. The AC believes the reviewer pointed out the overly narrow focus of the paper (i.e., main motivation: intra-modal alignment of CLIP image (or text) embedding space, application: few-shot image classification), and considers the comment valid as CLIP embeddings have been used in numerous applications so far; the impact of tackling the intra-modal misalignment issue in the image domain has not been investigated on other applications of CLIP embeddings. The authors presented additional experimental results but all of them are for intra-modal retrieval tasks, i.e., image-image retrieval and text-text retrieval, which failed to demonstrate the advantages of CLIP embeddings.*
	- *The other three reviewers (WGAS, hnwV, and ay5F) share a similar concern: evaluation is restricted only to image classification with a single backbone architecture and a single data modality. The authors conducted additional experiments for image-image / text-text retrieval and dataset transfer. However, as aforementioned, the tasks are all still limited to retrieval and they are not sufficient to demonstrate the versatile use of CLIP embeddings. Also, there is no experimental result with different backbones reported in the rebuttal.*
- Potential reliance on the custom prompts (JR1v, WGAS)
	- *The authors conducted additional experiments with different text prompts. With the results, the authors argued that the proposed method is not sensitive to the text prompts since it achieved strong performance even with the simple and generic prompts like "a photo of X," even significantly outperforming baselines. However, the AC found that the overall trends do not sufficiently support the authors' argument; the results of "a photo of X" does not look sufficiently strong and even inferior to many of the previous work. Also, considering the gap between the proposed method and the competitors in the first curve plot of Figure 5, the performance degradation by the use of simpler text prompts look significant. The authors should more carefully discuss the new results with detailed quantitative analysis to assuage the concern, but failed to do it. The AC thus considers the concerns only partially resolved.*

[Concerns that have been addressed successfully]
- Issues on conceptual illustration in Figure 2 (JR1v)
	- *The authors reflected the reviewer's comments to refine the figure. They also successfully addressed a concern on the validity of the conceptual illustration using cat and dog examples with established experimental results of previous work. The AC thus considers the issues on Figure 2 have been well resolved.*
- Incremental novelty (JR1v)
	- *The reviewer stated that the proposed method is a direct application of an existing model for image generation that creates a text embedding for images to CLIP-based few-shot classification. The authors argued that transferring the insight across distinct problem domains is a non-trivial contribution, and that the proposed method introduces a new supervised training strategy including new loss functions and bias terms that have not been investigated in the previous work. The AC found the authors' response satisfactory and thus considers the concern well resolved.*
- Lack of theoretical justification (WGAS)
	- *Reviewer WGAS asked a theoretical basis for the sufficiency of a single linear mapping. The response from the authors explains that the final text embedding of CLIP is computed by applying a linear projection to the EOS token and thus the pseudo inverse of the linear projection allows to bridge text and image features. The AC did not thoroughly examine the response line-by-line, but the overall idea sounds valid.*
- Marginal performance gain under more shots (hnwV)
	- *This is quite a natural phenomenon, and the authors' response successfully clarifies it.*
- Lack of detailed analysis on the quantitative results (hnwV)
	- *This has been successfully addressed by additional empirical analysis and evaluation in the rebuttal.*
- Lack of versatility (ay5F)
	- *Reviewer ay5F pointed out that the proposed method has been applied only to CLIP, damaging the versatility and consequently the future impact of the proposed method. The authors applied their method to SLIP and reported the results in the rebuttal. The AC thus considers that this concern has been addressed to some extent.*

**Reviewer Scores:**

While the reviewer hnwV recommended a borderline accept, they explicitly noted a low familiarity with the research topic (Confidence: 1) in both their initial review and the final comment. Similarly, the other positive reviewer WGAS expressed moderate confidence (Confidence: 3). In contrast, the two negative reviewers demonstrated greater certainty in their assessments, with higher confidence levels in their initial ratings (Confidence: 4). In addition, some critical concerns have not been successfully addressed by the rebuttal and the revision. The AC thus expects that, if the discussion phase has continued, the two positive reviewer may downgrade their ratings while the negative ones keep their ratings.

---

### Decision · Program_Chairs · 2026-01-26

Reject